# Latent-Predictive Empowerment: Measuring Empowerment without a Simulator

## Abstract

Empowerment has the potential to help agents learn large skillsets, but is not yet a scalable solution for training general-purpose agents. Recent empowerment methods learn large skillsets by maximizing the mutual information between skills and states, but these approaches require a model of the transition dynamics, which can be challenging to learn in realistic settings with high-dimensional and stochastic observations. We present an algorithm, Latent-Predictive Empowerment (LPE), that can compute empowerment in a more scalable manner. LPE learns large skillsets by maximizing an objective that under certain conditions has the same optimal skillset as the mutual information between skills and states, but our objective is more tractable to optimize because it only requires learning a simpler latent-predictive model rather than a full simulator of the environment. We show empirically in a variety of settings, includes ones with high-dimensional observations and highly stochastic transition dynamics, that our empowerment objective learns similar-sized skillsets as the leading empowerment algorithm, which assumes access to a model of the transition dynamics, and outperforms other model-based approaches to empowerment.

## 1 Introduction

Empowerment offers an intuitive approach for training agents to have large skillsets. In an empowerment-based approach, the empowerment for a variety of states is first computed, in which the empowerment of a state measures the size of the largest skillset in that state (Klyubin et al., 2005; Salge et al., 2013). The state empowerment values are then used as a reward in a Reinforcement Learning (Sutton & Barto, 1998) setting, encouraging agents to take actions that grow the size of their skillsets (Klyubin et al., 2008; Jung et al., 2012; Mohamed & Rezende, 2015).

The main roadblock to implementing the empowerment-based approach to training generalist agents is that there is not yet a scalable way to compute the empowerment of a state. Recent empowerment approaches seek to learn the most diverse skillset in a state by searching for the skillset (e.g., a skill-conditioned policy) with the largest lower bound to the mutual information between skills and states (Gregor et al., 2016; Eysenbach et al., 2018; Achiam et al., 2018; Lee et al., 2019; Choi et al., 2021; Strouse et al., 2021; Levy et al., 2024), which measures skillset diversity by capturing how distinct the skills are from one another in terms of the states they target. The problem with this approach is that it requires an infeasible amount of interaction with the environment prior to each update to the skillset. To estimate the mutual information lower bound for a single skillset in a single state, many skills need to be executed in the environment from the state under consideration to obtain the resulting skill-terminating states. But because empowerment seeks to find the most diverse skillset across a distribution of states, these tuples of skills and states need to be collected for many skillsets (e.g., skill-conditioned policies with small differences from the policy) starting from many states. Because this amount of interaction prior to each update to the skillset is intractable, recent empowerment approaches assume the agent has access to a model of the transition dynamics (i.e., a simulator of the environment) (Eysenbach et al., 2018; Gu et al., 2021; Levy et al., 2023; 2024). But this is not a scalable assumption because a model of the transition dynamics is typically not available and can be intractable to learn in settings with high-dimensional and stochastic observations.

We present a more scalable approach for measuring empowerment, *Latent-Predictive Empowerment (LPE)*. LPE measures the diversity of a skillset using the difference of two terms: (i) the mutual in-

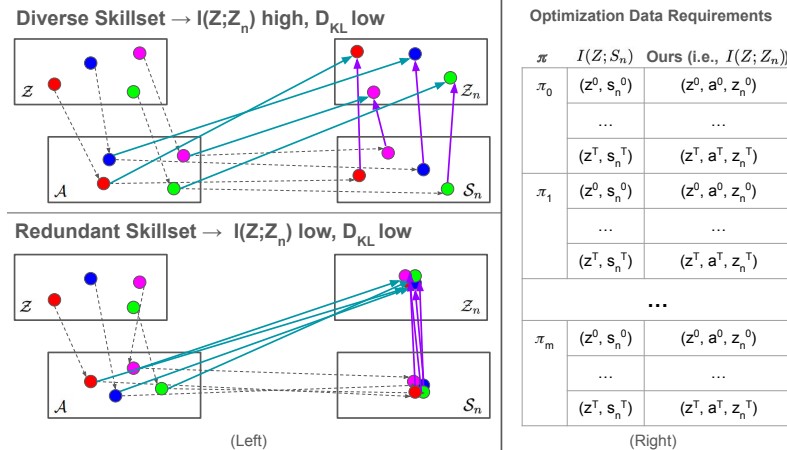

Figure 1: (Left) Illustration of the latent-predictive model and state encoding distributions for both diverse and redundant skillsets. The different colored circles represent different tuples of (skills, open loop action sequences, skill-terminating states, and skill-terminating latent representations) generated by a skillset. For a diverse skillset in which different skills target different states, the latent-predictive model (teal arrows), which maps actions to latent states, can output unique latent states that match the output of the state encoding distribution (purple arrows), which maps skill-terminating states to latent vectors. This produces a high overall diversity score because the mutual information between skills and latent states, $I(Z; Z_n)$, is high because different skills target different latent states, and the KL divergence between the latent-predictive model and state encoding distribution is low. On the other hand, for redundant skillsets in which different skills target the same states, the latent-predictive model may map different actions to the same latent vector yielding a low overall diversity score because $I(Z; Z_n)$ is low. (Right) Comparison of the data required to optimize (i) $I(Z; S_n)$, the mutual information between skills and states, and (ii) our objective. For each candidate skillset $\pi_i$ (left column), $I(Z; S_n)$ may require $T$ tuples of (skill $z$, skill-ending state $s_n$), which requires access to a simulator of the environment. On the other hand, most of the required data for our objective consists of the (skill $z$, action sequence $a$, latent representation $z_n$) tuples needed to estimate $I(Z; Z_n)$ for all candidate skillsets, which only requires learning a latent-predictive model.

formation between skills and latent representations of states generated by a latent-predictive model and (ii) an average KL divergence that measures the mismatch between the the latent-predictive model and a state encoding distribution. The objective provides an intuitive way to measure skillset diversity. The mutual information term measures the number of different actions that a skillset executes, and the KL divergence term penalizes redundant actions that achieve the same terminating states as other actions in the skillset. Figure 1 (Left) describes the latent-predictive and state encoding distributions and visualizes what both distributions can look like for diverse and redundant skillsets. Our objective for measuring skillset diversity offers a more scalable way to learn diverse skillsets because most of the data that is needed to optimize the objective consists of tuples of (skills, open loop action sequences, and skill-terminating latent representations), which are used to estimate the mutual information between skills and latent states for different skillsets. To generate this data only requires a latent-predictive model, which can be significantly more tractable to learn than a full simulator because it operates in a lower dimensional latent space and can be implemented as a simple distribution such as a diagonal gaussian. Figure 1 (Right) compares the data required for maximizing (a) the mutual information between skills and states and (b) our objective for measuring skillset diversity. In addition, although our objective is different than the mutual information between skills and states, we show that it is still a principled replacement for this mutual information because it has the same optimal skillset under certain conditions.

Our experiments in a series of domains, including settings with stochastic and high-dimensional observations, demonstrate that our approach can learn large skillsets, matching the skillset sizes achieved by the leading empowerment algorithm that assumes access to a simulator of the environment. Our algorithm also significantly outperforms other model-based approaches to empowerment that learn some type of model of the environment. To our knowledge, Latent-Predictive Empow-

erment is the first unsupervised skill learning method to learn large skillsets in stochastic settings without a simulator.

## 2 BACKGROUND

### 2.1 SKILLSET MODEL AND EMPOWERMENT

We model an agent's skillset in a state using a probabilistic graphical model defined by the tuple $(\mathcal{S}, \mathcal{A}, \mathcal{Z}, T, \phi, \pi)$. $\mathcal{S}$ is the space of states; $\mathcal{A}$ is the space of actions; $\mathcal{Z}$ is the space of skills; $T$ is the transition dynamics distribution $T(s_{t+1}|s_t, a_t)$ that provides the probability of a state given the prior state and action. The transition dynamics are assumed to be conditionally independent of the history of states and actions (i.e., $T(s_{t+1}|s_t, a_t) = T(s_{t+1}|s_0, a_0, \ldots, s_t, a_t)$). The remaining distributions $\phi$ and $\pi$ are the learnable distributions in a skillset. $\phi$ represents the distribution over skills $\phi(z|s_0)$ given a skill start state $s_0$. $\pi$ represents the skill-conditioned policy $\pi(a_t|s_t, z)$ that provides the distribution over primitive actions given a state $s_t$ and skill $z$. Assuming each skill consists of $n$ primitive actions, the full joint distribution of a skill and a trajectory of actions and states $(z, a_0, s_1, \ldots, a_{n-1}, s_n)$ conditioned on a particular start state $s_0$ and skillset defined $\phi$ and $\pi$ is given by $p(z, a_0, s_1, \ldots, a_{n_1}, s_n|s_0, \phi, \pi) = \phi(z|s_0)\pi(a_0|s_0, z)p(s_1|s_0, a_0)\ldots\pi(a_{n-1}|s_{n-1}, z)p(s_n|s_{n-1}, a_{n-1})$. Note that this definition is for closed loop skills. Skillsets can also use open loop skills, in which the skill-conditioned policy would be defined defined by the distribution $\pi(a|s_0, z)$. The output of the open loop skill-conditioned policy $a$ is a concatenation of $n$ primitive actions (i.e., $a = [a_0, \ldots, a_{n-1}]$). The joint distribution for a skillset containing open loop skills is the same as for closed loop skills except there is only one sample taken from the skill-conditioned policy, which includes all $n$ primitive actions.

In this paper, we measure the diversity of a skillset defined by $\phi$ and $\theta$ using the mutual information between the skill random variable $Z$ and the skill-terminating state random variable $S_n$, $I(Z; S_n|s_0, \phi, \pi)$. This mutual information measures the number of distinct skills in a skillset, in which a skill is distinct if it targets a set of states not targeted by other skills in the skillset. $I(Z; S_n|s_0, \phi, \pi)$ is defined

$$I(Z; S_n|s_0, \phi, \pi) = H(Z|s_0, \phi, \pi) - H(Z|s_0, \phi, \pi, S_n) \tag{1}$$

$$= \mathbb{E}_{z \sim \phi(z|s_0), s_n \sim p(s_n|s_0, \pi, z)}[\log p(z|s_0, \phi, \pi, s_n) - \log p(z|s_0, \phi)]. \tag{2}$$

Per line 1, the diversity of a skillset grows when there are more skills in a skillset (i.e., higher skill distribution entropy $H(Z|s_0, \phi, \pi)$) and/or the skills become more distinct (i.e., the conditional entropy $H(Z|s_0, \phi, \pi, S_n)$ shrinks).

The empowerment of a state is the maximum mutual information with respect to all possible $(\phi, \pi)$ skillsets:

$$\mathcal{E}(s) = \max_{\phi, \pi} I(Z; S_n|s, \phi, \pi). \tag{3}$$

That is, the empowerment of a state measures the size of the largest possible skillset in that state. Note that this use of empowerment, in which mutual information is maximized to find the most largest possible skillset in a range of states, enables a different use of empowerment, which is as a reward for decision-making. In this other use case of empowerment that is also common in the literature, agents are rewarded for taking actions that grow the size of their skillsets (Klyubin et al., 2008; Jung et al., 2012; Mohamed & Rezende, 2015).

### 2.2 SKILLSET EMPOWERMENT

A leading algorithm for computing empowerment is Skillset Empowerment (Levy et al., 2024), which measures a variational lower bound on empowerment, $\tilde{E}(s_0)$, defined as follows:

$$\tilde{\mathcal{E}}(s_0) = \max_{\phi, \pi} \tilde{I}(Z; S_n|s_0, \phi, \pi), \tag{4}$$

$$\tilde{I}(Z; S_n|s_0, \phi, \pi) = \mathbb{E}_{z \sim \phi(z|s_0), s_n \sim p(s_n|s_0, \pi, z)}[\log q_{\psi^*}(z|s_0, \phi, \pi, s_n) - \log \phi(z|s_0)],$$

$$\psi^* = \arg\min_{\psi} D_{KL}(p(z|s_0, \phi, \pi, s_n)||q_\psi(z|s_0, \phi, \pi, s_n)). \tag{5}$$

Skillset Empowerment measures a tighter lower bound on empowerment than prior work (Gregor et al., 2016; Eysenbach et al., 2018; Achiam et al., 2018; Lee et al., 2019; Choi et al., 2021; Strouse et al., 2021) because, for any candidate $(\phi, \pi)$ skillset, it learns a tighter variational lower bound $\tilde{I}(Z; S_n|s_0, \phi, \pi)$ on the true mutual information $I(Z; S_n|s_0, \phi, \pi)$ as a result of (i) conditioning the variational posterior, $q_\psi(z|s_0, \phi, \pi, s_n)$, on the $(\phi, \pi)$ skillset distributions and then (ii) training the variational posterior $q_\psi(z|s_0, \phi, \pi, s_n)$ for a candidate $(\phi, \pi)$ skillset to match the true posterior $p(z|s_0, \phi, \pi, s_n)$ of the candidate skillset. As a result of this tighter lower bound on empowerment, Skillset Empowerment was the first unsupervised skill learning algorithm to learn large skillsets in domains with stochastic and high-dimensional observations. Skillset Empowerment maximizes the variational mutual information $\tilde{I}(Z; S_n|s_0, \phi, \pi)$ with respect to the skillset distributions $\phi$ and $\pi$ using a particular actor-critic architecture. We will be using the same actor-critic architecture in our approach, so we review this architecture in section A of the Appendix.

The problem with Skillset Empowerment is that it is not a scalable approach for measuring the empowerment of a state because it assumes a model of the transition dynamics, $p(s_{t+1}|s_t, a_t)$, is either provided or learned. But this is not a practical assumption in real world settings where a simulator of the environment is typically not available and is too difficult to learn because it is hard to predict high-dimensional and stochastic future observations. Skillset Empowerment requires a model of the transition dynamics because of the large number of (skill $z$, skill-terminating state $s_n$) tuples needed to optimize the objective. In order to estimate the variational mutual information $\tilde{I}(Z; S_n|s_0, \phi, \pi)$ for a single candidate skillset $(\phi, \pi)$, Skillset Empowerment requires many $(z, s_n)$ tuples to learn the parameters $\psi^*$ for the variational posterior. This is because in practice the KL divergence minimization objective provided in equation 5 is implemented as a maximum likelihood objective: $\mathbb{E}_{z\sim\phi(z|s_0), s_n\sim p(s_n|s_0, \pi, z)}[\log q_\psi(z|s_0, \phi, \pi, s_n)]$, which requires $(z, s_n)$ samples to find the best fitting variational posterior. But in order to learn the empowerment of a state, or the maximum mutual information with respect to $(\phi, \pi)$, the variational mutual lower bound needs to be estimated for a large number of combinations of skill distributions $\phi(z|s_0)$ and skill-conditioned policies $\pi(a_t|s_t, z)$. In Skillset Empowerment specifically, the variational lower bound on mutual information needs to be computed for small changes to each of the potentially thousands of parameters that make up $\pi$. Obtaining the required $(z, s_n)$ tuples for a large number of $(\phi, \pi)$ skillsets in an online fashion is not practical, which is why Skillset Empowerment requires access to a model of the transition dynamics $p(s_{t+1}|s_t, a_t)$.

## 3 LATENT-PREDICTIVE EMPOWERMENT

We introduce a new algorithm, Latent-Predictive Empowerment (LPE), that can measure the empowerment of a state in a more scalable manner. The key component of our algorithm is our objective for learning diverse skill-conditioned policies. Instead of maximizing the mutual information between skills and states with respect to the skill-conditioned policy, we maximize an alternative objective that has the same optimal skillset under certain conditions, but is also more tractable to maximize because it only requires learning a latent-predictive model rather than a full simulator of an environment. We maximize skillset diversity using the same actor-critic structures as used by Skillset Empowerment, which is reviewed in Appendix section A.

### 3.1 TRAINING OBJECTIVE FOR SKILL-CONDITIONED POLICY ACTOR

In Latent-Predictive Empowerment, the objective used to train the skill-conditioned policy actor so that it outputs diverse skill-conditioned policies $\pi$ given a skill start state $s_0$ and skill distribution $\phi$ is:

$$\mathcal{E}_{LPE,\pi}(s_0, \phi) = \max_\pi J(s_0, \phi, \pi), \tag{6}$$

$$J(s_0, \phi, \pi) = \tilde{I}(Z; Z_n|s_0, \phi, \pi) - \mathbb{E}_{(a,s_n)\sim p(a,s_n|s_0,\pi)}[D_{KL}(p_\xi(z_n|s_0, \phi, \pi, a)||p_\eta(z_n|s_0, \phi, \pi, s_n))],$$

$$\tilde{I}(Z; Z_n|s_0, \phi, \pi) = \mathbb{E}_{z\sim\phi(z|s_0), a\sim\pi(a|s_0,z), z_n\sim p_\xi(z_n|s_0, \phi, \pi, a)}[\log q_\phi(z|s_0, \phi, \pi, z_n) - \log \phi(z|s_0)].$$

That is, for a given skill start state $s_0$ and skill distribution size $\phi$, Latent-Predictive Empowerment seeks to find the most diverse skill-conditioned policy $\pi$, in which skillset diversity is measured by $J(s_0, \phi, \pi)$. $J(s_0, \phi, \pi)$ consists of the difference of two terms, which we describe next.

### 3.1.1 Intuitive Skillset Diversity Objective

The first term, $\tilde{I}(Z; Z_n | s_0, \phi, \pi)$, in LPE's objective for measuring skillset diversity is the variational lower bound on the mutual information between skills and latent state representations, in which the latent state is generated by the latent-predictive model $p_\xi(z_n | s_0, \phi, \pi, a)$, which maps open loop action sequences $a$ to the latent vector $z_n$ for the given skill start state $s_0$ and skillset distributions $\phi$ and $\pi$. This is a variational lower bound on mutual information because the variational posterior $q_\psi(z | s_0, \phi, \pi, z_n)$ replaces the intractable true posterior $p(z | s_0, \phi, \pi, z_n)$ (Barber & Agakov, 2003). Given that $\tilde{I}(Z; Z_n | s_0, \phi, \pi)$ is a lower bound on the mutual information between skills and actions $I(Z; A | s_0, \phi, \pi)$ via the data processing inequality (Cover & Thomas, 2006), the contribution that the $\tilde{I}(Z; Z_n | s_0, \phi, \pi)$ term makes to measuring skillset diversity is that it measures how many different actions the $(\phi, \pi)$ skillset executes in state $s_0$. The more unique open loop action sequences executed by the $(\phi, \pi)$, regardless of the states they target, the higher the $\tilde{I}(Z; Z_n | s_0, \phi, \pi)$ can be. Note that when trained to maximize $\tilde{I}(Z; Z_n | s_0, \phi, \pi)$, the latent-predictive model $p_\xi(z_n | s_0, \phi, \pi, a)$ is encouraged to output unique latent vectors $z_n$ for each open loop action sequence $a$.

The second term in the skillset diversity objective is an average KL divergence between the latent-predictive model $p_\xi(z_n | s_0, \phi, \pi, a)$ and the state encoding distribution $p_\eta(z_n | s_0, \phi, \pi, s_n)$, which encodes skill-terminating states $s_n$ to latent states $z_n$ for a given skill start state $s_0$ and $(\phi, \pi)$ skillset. The KL divergence is averaged over the different (open loop action sequence $a$, skill-terminating state $s_n$) generated by the $(\phi, \pi)$ skillset under consideration. The contribution of this term to measuring skillset diversity is to penalize the skillset for any skills that the $\tilde{I}(Z; Z_n | s_0, \phi, \pi)$ term had counted as unique because they output different actions but actually target the same terminating state. For instance, if there are two distant skills $z$ that execute different actions $a$ that target the same state $s_n$, and the latent-predictive model $p_\xi$ assigns two distant latent states $z_n$ for the different actions, then the KL divergence will lower the diversity score because the state encoding distribution will need to take on a more entropic distribution to cover the different latent states output by the latent-predictive model in order to minimize the KL divergence. Note that when the latent-predicted model $p_\xi$ and the state encoding distribution $p_\eta$ are jointly trained to minimize this KL divergence, they are encouraged to output similar distributions.

The two terms together provide an intuitive way to measure skillset diversity, in which skillsets that execute more distinct actions that target distinct grouping of states are assigned higher diversity scores. In regards to the form the the latent-predictive $p_\xi$, state encoding $p_\eta$, and variational posterior $q_\phi$ distributions take when they are are jointly trained to maximize the diversity score for a particular $(\phi, \pi)$ skillset, the latent-predictive model is encouraged to output latent states that both (a) match the output of the state encoding distribution (decreasing the KL divergence) and (b) are unique so that they can be decoded back to the original skill via the variational posterior $q_\phi(z | s_0, \phi, \pi, z_n)$ (increasing $\tilde{I}(Z; Z_n)$). For diverse skillsets in which different actions target different states, the distributions can take this form, as illustrated in Figure 5 of the Appendix.

### 3.1.2 Tractable Data Requirements

Next we discuss the data required to maximize LPE skillset diversity objective with respect to the skill-conditioned policy $\pi$. Because we will use the same actor-critic optimization architecture as Skillset Empowerment in which a critic is trained for each parameter of the skill-conditioned policy $\pi$, we will need to measure the diversity of a large number of skillsets that contain some changes to each of the $\pi$ parameters of the skill-conditioned policy. To measure the diversity of a single $(\phi, \pi)$ skillset (i.e., optimize the $J(s_0, \phi, \pi)$ objective with respect to the latent-predictive model, state encoding distribution, and variational posterior), (i) tuples of (skills $z$, open loop action sequences $a$, and skill-terminating latent states $z_n$) are needed to optimize the variational mutual information $\tilde{I}(Z; Z_n | s_0, \phi, \pi)$ and (ii) transition tuples of (skill start state $s_0$, action sequence $a$, skill-terminating state $s_n$) generated by the $(\phi, \pi)$ are needed to optimize the KL divergence between the latent-predictive model and the state encoding distribution.

Obtaining this data for a large number of skillsets is significantly more tractable then acquiring the data needed to optimize the variational lower bound on the mutual information between skills and states $\tilde{I}(Z; S_n | s_0, \phi, \pi)$ as is done by Skillset Empowerment. $\tilde{I}(Z; S_n | s_0, \phi, \pi)$ required a large number of $(z, s_n)$ tuples which needed a simulator of the environment to generate the states $s_n$,

which can be high-dimensional and stochastic. On the other hand, the $(z, a, z_n)$ tuples needed to optimize the $\tilde{I}(Z; Z_n|s_0, \phi, \pi)$ term only requires a latent-predictive model, which is more feasible to train because it predicts lower dimensional latent states and the latent-predictive model can take the form of simple distribution like a diagonal gaussian. In addition, the needed transition data $(s_0, a, s_n)$ can be mostly sampled from a replay buffer of online transition data. In the LPE algorithm, we will assume the agent, in between updates to its skillset, interacts with the environment by sampling skills $z \sim \phi(z|s_0)$ from its skillset, greedily executing its skill-conditioned policy $\pi(a|s_0, z)$, and then storing the $(s_0, a, s_n)$ transitions that occur. In section C of the Appendix we discuss how LPE responds to skillsets that execute new actions that are not in the replay buffer and why this helps LPE explore new skillsets.

### 3.1.3 PRINCIPLED REPLACEMENT FOR $I(Z; S_n|s_0, \phi, \pi)$

The skillset diversity objective used in equation 6 is a principled replacement for the mutual information between skills and states $I(Z; S_n|s_0, \phi, \pi)$ because under some relatively reasonable assumptions they have the same maximum with respect to the skill-conditioned policy $\pi$ (see section E for proof and additional commentary on the assumptions). The assumptions include (i) there exists some finite maximum posterior for the relevant true and variational posteriors and that (ii) there exists a $(\phi, \pi)$ skillset such that $\pi$ produces maximum variational posteriors $q_\psi$. In practice, the second assumption is more realistic for small skill distributions $\phi$ because for large distributions there may not be enough states that can be targeted to produce only tight posteriors. The proof makes use of the following connection between the skillset diversity objective $J(s_0, \phi, \pi)$ and the mutual information between skills and states, $I(Z; S_n|s_0, \phi, \pi)$. In the first step of this connection, we note that the LPE skillset diversity objective $J(s_0, \phi, \pi)$ is a lower bound of the following objective (see Appendix section D for proof)

$$I_J(s_0, \phi, \pi) = H(Z|s_0, \phi) + \log(\mathbb{E}_{s_n \sim \phi(z|s_0), z_n \sim p_\eta(z_n|s_0, \phi, \pi, z)}[p(z|s_0, \phi, \pi, z_n)]). \tag{7}$$

Thus, by maximizing the skillset diversity objective $J(s_0, \phi, \pi)$ with respect to $\pi$ (and $p_\xi, p_\eta$, and $, q_\phi$), LPE is learning $(\phi, \pi)$ skillsets with larger true posterior distributions $p(z|s_0, \phi, \pi, z_n)$, meaning that agents are learning skillsets with more distinct skills "packed" inside them. Next, we note that the $I_J(s_0, \phi, \pi)$ objective is an upper bound of the the mutual information between skills and latent representations $I(Z; Z_n|s_0, \phi, \pi)$, in which the latent representation $z_n \sim p_\eta(z_n|s_0, \phi, \pi, s_n)$ is sampled from the state encoding distribution. The inequality is due to Jensen's Inequality as $I_J$ has an log of an expectation over posteriors term while $I(Z; Z_n|s_0, \phi, \pi)$ has an expectation of the log of the posteriors. We complete the connection by noting that $I(Z; Z_n|s_0, \phi, \pi)$ is a lower bound to $I(Z; S_n|s_0, \phi, \pi)$ using the data processing inequality. In the proof, we show that for certain $(\phi, \pi)$ skillsets, these inequalities become equalities and the same $\pi$ can optimize both $J(s_0, \phi.\pi)$ and $I(Z; S_n|s_0, \phi, \pi)$.

### 3.1.4 PRACTICAL IMPLEMENTATION OF $\pi$ ACTOR-CRITIC

LPE learns diverse skill-conditioned policies $\pi$ for a variety of skill start state $s_0$ and skill distribution $\phi$ combinations using a similar actor-critic architecture to the one used by Skillset Empowerment, which we review in section A of the Appendix. The actor $f_\lambda$ will take as input a $(s_0, \phi)$ tuple and output a skill-conditioned policy parameter vector $\pi$. The parameter-specific critic $Q_{\omega_i}$ for $i = 0, \ldots, |\pi| - 1$ will measure the $J(s_0, \phi, \pi)$ diversity of skillsets defined by $(s_0, \phi, \pi_i)$ tuples, in which $\pi_i = f_\lambda(s_0, \phi)$ except for the $i$-th parameter which can take on noisy values. We detail the objectives using for training the actor and critics in section F of the Appendix.

### 3.2 TRAINING OBJECTIVE FOR SKILL DISTRIBUTION ACTOR

We train the skill distribution actor $f_\mu$ actor, which outputs a distribution over skills $\phi$ for a given $s_0$, to maximize the variational mutual information objective $\tilde{I}(Z; Z_n|s_0, \phi, \pi = f_\lambda(s_0, \phi))$, in which $z_n \sim p_\xi(z_n|s_0, \phi, \pi = f_\lambda(s_0, \phi), a)$ is sampled from a latent-predictive model, which has been trained to match the state encoding distribution $p_\eta(z_n|s_0, \phi, \pi = f_\lambda(s_0, \phi), s_n)$. As discussed previously, $\tilde{I}(Z; Z_n|s_0, \phi, \pi = f_\lambda(s_0, \phi))$ offers a principled substitute for the mutual information between skills and states $I(Z; S_n|s_0, \phi, \pi)$, particularly for relatively small $\phi$. Note that we do not use the same latent-predictive model that was trained during the skill-conditioned policy

actor-critic update, but instead train a new latent-predictive model to minimize the KL divergence between the state-encoding distribution and the new latent-predictive model. We train a new model because for relatively larger values of $\phi$, there could be a scenario in which a $\pi$ is learned that trades off artificially high $\tilde{I}(Z; Z_n | s_0, \phi, \pi)$ (i.e., redundant skills are treated as unique skills) for lower $D_{KL}(p_\xi || p_\eta)$, which would mean the learned latent-predictive model is not accurate. Although the latent-predictive models learned in the $\pi$ actor-critic were diagonal gaussian, we implement the latent-predictive model in the $\phi$ update using the more expressive Variational Autoencoder (VAE) (Kingma & Welling, 2022). The objective for training the VAE is provided in section G of the Appendix.

The objective functions used to train the $\phi$ actor and critic are provided in section H of the Appendix. The full LPE procedure is provided in Algorithm 1.

---

**Algorithm 1** Latent-Predictive Empowerment (LPE)

   **repeat**
       Greedily execute skillset in environment and store $(s_0, a, s_n)$ transitions in buffer
       Update skill-conditioned policy $\pi$ actor-critic (see equations 11 - 14)
       Update VAE-based latent-predictive model (see equation 15)
       Update skill distribution $\phi$ actor-critic (see equations 16-18)
   **until** convergence

---

### 3.3 LIMITATIONS

The main limitation of Latent-Predictive Empowerment is that it can be limited to measuring only short term empowerment because of the use of open loop skills. The inability to adjust a policy makes it difficult to target specific states over longer time horizons, particularly in domains with realistic levels of randomness. As a result, LPE can be a poor way to measure longer term empowerment. Future work can investigate how a longer term empowerment can be computed from the short term empowerment measured by LPE.

## 4 EXPERIMENTS

### 4.1 ENVIRONMENTS

We test LPE and a group of baselines on the same five domains that were used in Skillset Empowerment. Along the dimensions of stochasticity and the dimensionality of observations, these environments are complex because all but one have highly stochastic transition dynamics and some include high-dimensional state observations. On the other hand, in terms of the dimensionality of the underlying state space not visible by the agent, all domains have simple, low-dimensional underlying state spaces. Stochastic domains are used because general purpose agents need to be able to build large skillsets in environments with significant randomness, and there are already effective algorithms for learning skills in deterministic settings (e.g., unsupervised goal-conditioned RL methods). Low-dimensional underlying state environments are used in order to limit the parallel compute needed to implement both Skillset Empowerment and Latent-Predictive Empowerment because both approaches require a significant amount of compute to train the parameter-specific critics in parallel even for simple settings. Section I in the Appendix provides information on the number of GPUs used in the experiments.

The first two experiments are built in a stochastic four rooms setting. In the navigation version of this setting, a two-dimensional point agent executes 2D (i.e., $(\Delta x, \Delta y)$) actions in a setting with four separated rooms. After each action is complete, the agent is moved randomly to the corresponding point in one of the four rooms. In the pick-and-place version of this setting, there is a two-dimensional object the agent can move around if the agent is within a certain distance. The abstract skills agents can learn in these domains are to target $(x, y)$ offset positions from the center of a room for the agent (and for the object in the pick-and-place version). The other two stochastic environments are built in an RGB-colored QR code domain, in which a 2D agent moves within a lightly-colored QR code where every pixel of the QR code changes after each action. The state observations are 432 dimensional (12x12x3 images). We also created a pick-and-place version of this

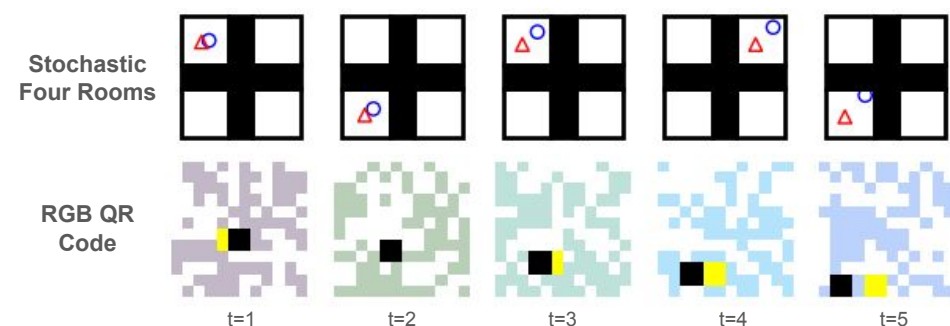

Figure 2: Sample skill sequences in the pick-and-place versions of the Stochastic Four Rooms and RGB QR Code domains. In top row, the blue circle agent executes a skill to move away from red triangle object. In bottom row, the black square agent carries the yellow object to bottom of room.

task, in which the agent can move around an object provided the object is within reach. The abstract skills to learn in these domains are again to target $(x, y)$ locations for the agent (and the object in the pick-and-place version). Image sequences showing executed skills in the pick-and-place versions of the stochastic tasks are shown in Figure 2. We also applied the algorithms to the continuous mountain car domain (Towers et al., 2024) to test whether agents can learn skills to target states containing both positions and velocities. In addition, to test LPE in a setting with a larger underling state space, we implemented an 8-dim room environment in which states and actions are 8-dim, and the dynamics simply consist of the state dimensions changing by the amounts listed in the action. Additional details for these domains are provided in section J of the Appendix.

Given our goal of a more scalable way to measure the empowerment of a state, we evaluate the performance of LPE and the baselines by the size of the skillsets they learn in each domain. We measure the size of the skillsets using the variational mutual information $\tilde{I}(Z; S_n | s_0, \phi, \pi)$ from a single start state $s_0$. In this paper, we are not assessing performance on downstream tasks, in which, for instance, a hierarchical agent needs to learn a higher level policy that executes skills from the learned skillsets to maximize some reward function. However, in section K of the Appendix we describe how it is simple to implement such hierarchical agents that use the $(\phi, \pi)$ LPE skillsets as a temporally extended action space.

## 4.2 BASELINES

We compare LPE to three versions of Skillset Empowerment. The first version is regular Skillset Empowerment, in which the agent is given access to the model of transition dynamics. Levy et al. (2024) showed that Skillset Empowerment is able to learn large skillsets in all domains while both Variational Intrinsic Control (Gregor et al., 2016), an empowerment-based skill learning algorithm similar to Diversity Is All You Need (Eysenbach et al., 2018), and Goal-Conditioned RL were unable to learn meaningful skillsets. In the second version, the Skillset Empowerment agent learns a model of the transition dynamics $p(s_{t+1} | s_t, a_t)$ using a VAE (Kingma & Welling, 2022) generative model. We expect this agent to struggle in the stochastic settings because it is challenging to learn simulators in these domains, which in turn means the agent may struggle to accurately measure the diversity of a skillset. Learning a simulator in stochastic four rooms is difficult because the agent's next location occurs in the same offset location in any of the four rooms and it is difficult for VAE's to learn disjoint distributions. Further, learning a perfect simulator in the RGB QR code domains in which the agent needs to predict the next QR code is not feasible due to the number of RGB-colored QR code combinations.

In the third version of Skillset Empowerment, the agent learns a latent-predictive model using a BYOL-Explore objective (Guo et al., 2022), which is a leading method for learning latent-predictive models. Similar to other bootstrapping methods (Grill et al., 2020; Assran et al., 2023; Bardes et al., 2024), BYOL-Explore trains a latent-predictive model $p_\xi(z_n | s_0, a)$ to match a state encoding distribution $p_\eta(z_n | s_n)$, in which the parameters of the state encoding distribution are updated as an exponential moving average of the latent-predictive model parameters: $\eta \leftarrow \alpha\eta + (1-\alpha)\xi$. We also

Table 1: Average (+Std) Learned Skillset Size over 5 random seeds (units: nats)

| Algorithm | S4R Nav | S4R PP | QR Code Nav | QR Code PP | Mtn. Car |
|---|---|---|---|---|---|
| LPE | $6.5 \pm 0.4$ | $8.6 \pm 0.5$ | $4.2 \pm 0.1$ | $6.5 \pm 0.4$ | $6.4 \pm 0.4$ |
| SE | $5.1 \pm 0.3$ | $8.7 \pm 0.3$ | $3.5 \pm 0.1$ | $6.0 \pm 0.2$ | $5.3 \pm 0.3$ |
| SE+BYOL | $1.6 \pm 0.3$ | $2.4 \pm 0.3$ | $0.8 \pm 0.4$ | $1.6 \pm 0.3$ | $5.4 \pm 0.3$ |
| SE+VAE | $2.7 \pm 0.6$ | $1.8 \pm 0.7$ | $2.4 \pm 0.8$ | $3.2 \pm 0.7$ | $5.0 \pm 0.1$ |

expect this approach to struggle because it is susceptible to only maximizing a loose lower bound on the mutual information between skills and states. By training the latent-predictive model to match the state encoding distribution (i.e., minimize $D_{KL}(p_\eta(z_n|s_0,a)||p_\xi(z_n|s_0,a))$), this approach will be measuring the diversity of skillsets using the mutual information $I(Z; Z_n|s_0, \phi, \pi)$, in which $z_n$ is generated by the state encoding distribution $p_\eta(z|s_0, s_n)$. This mutual information is a lower bound on the mutual information between skills and states $I(Z; S_n|s_0, \phi, \pi)$ due to the data processing inequality, and the tightness of this bound depends on the state encoding distribution $p_\eta(z_n|s_0, s_n)$. If $p_\eta$ maps states different $s_n$ to different latent states $z_n$, then this bound can be tight, but otherwise this bound can be loose. The problem with BYOL is that it does not directly train the state-encoding distribution $p_\eta$ to output unique $z_n$ for different $s_n$. Instead, as a result of the exponential moving average update strategy, the output of the state-encoding distribution depends significantly on the initial parameter settings of $\eta$. If the initial setting of $\eta$ does not map different $s_n$ to different $z_n$, then $I(Z; Z_n|s_0, \phi, \pi)$ may be a loose bound on $I(Z; S_n|s_0, \phi, \pi)$, meaning the agent is not able to accurately measure the diversity of a skillset. In contrast, LPE does not have this issue because the state-encoding distribution is trained to match the latent-predictive model, which is also trained to maximize the mutual information $I(Z; Z_n|s_0, \phi, \pi)$, encouraging the latent-predictive model and the state encoding distribution to output unique $z_n$ for different inputs.

### 4.3 RESULTS

Table 1 shows the size of the skillsets learned by all algorithms in all domains except for the 8-dim underlying state domain where the agents learned an average skillset size of $15.9 \pm 0.6$ nats. Skillset size is measured with the variational mutual information $\tilde{I}(Z; S_n)$. Note that mutual information in measured on a logarithmic scale (in this case, nats) so the 8.6 nats of skills learned by LPE in the pick-and-place version of the Stochastic Four Rooms domain means that LPE learned $e^{8.6} \approx 5,400$ skills. The results of our experiments show that Latent-Predictive Empowerment can match the size of the skillsets learned by Skillset Empowerment despite (a) not having access to a simulator of the environment and (b) maximizing a different objective than $\tilde{I}(Z; S_n)$. In addition, neither of the Skillset Empowerment variants with learned models were able to learning meaningful skillsets in the stochastic domains, but were able to learn large skillsets in the deterministic continuous mountain car domain.

For additional evidence that LPE is able to learn large skills in all domains, we provide visualizations of the mutual information entropy terms (i.e., $H(S_n), H(S_n|Z), H(Z), H(Z|S_n)$) both before and after training in Figures 6-17 in the Appendix. The $H(S_n)$ visuals shows the skill-terminating states $s_n$ achieved by 1000 skills randomly sampled from the learned skill distribution. In all tasks, the skill-terminating states nearly uniformly cover the reachable state space. To show that this was not achieved by simply executing a policy that uniformly samples actions from the action space, in the center image we visualize $H(S_n|Z)$, which shows 12 skill-terminating states $s_n$ from four randomly selected skills from the skill distribution. In the stochastic settings, for instance, the $s_n$ generated by each skill $z$ target a specific $(x, y)$ offset location for the agent and an $(x, y)$ offset location for the object in the pick-and-place tasks, which is the correct abstract skill to learn. These visuals also visualize $H(Z)$ by showing the distribution over skills $\phi$ that takes the shape of a $d$-dimensional cube. Lastly, we visualize $H(Z|S_n)$ by showing four randomly selected skills $z$ and samples from the learned posterior $q_\psi(z|s_n)$. As expected for a diverse skillset in which different skills target different states, these samples of the posterior distribution tightly surround the original skill.

We note that searching across the space of $(\phi, \pi)$ skillsets for a skillset that targets a diverse distribution of skill-terminating states is not a trivial task in these domains. A skill-conditioned policy

that randomly executes actions would produce a zero mutual information skillset. A skillset that tried to maximize the mutual information $I(Z; A)$ (i.e., have each skill execute a different action) would also produce relatively low $I(Z; S_n)$ because among the space of open loop action sequences $a_0, a_1, \ldots, a_{n-1}$, many of these sequences target the same skill-terminating state $s_n$. In addition, the need to have the skillset fit a diagonal gaussian variational posterior $q_\psi(z|s_0, \phi, \pi, s_n)$ also makes the task challenging because a skillset in which distant skills $z$ target the same state $s_n$ can produce a low $\tilde{I}(Z; S_n)$ score because this would result in a high entropy variational posterior $q_\psi$. Instead, each small region of the skill distribution needs to target a distinct grouping of states $s_n$.

Moreover, our results show that the stochastic domains exposed the flaws in the Skillset Empowerment variants. Figure 18 in the Appendix shows how the VAE generative model often struggled to learn sufficiently accurate transition dynamics, which resulted in inaccurate skillset diversity measurements. For the BYOL variant, stochastic domains make it more likely that Skillset Empowerment will only be maximizing a loose bound on mutual information and thus not accurately measure skillset diversity. In stochastic settings where actions can produce a large number of different states, BYOL would need the initial parameters $\eta$ of the state encoding distribution to map most of these states $s_n$ to unique $z_n$, but this is unlikely.

## 5 RELATED WORK

There have been many prior works that have used empowerment to try to learn large skillsets. Early empowerment methods showed how mutual information between actions and states could be optimized in small settings with discrete state and/or action spaces (Klyubin et al., 2008; Salge et al., 2013; Jung et al., 2012). Several later works integrated variational inference techniques that enabled empowerment-based skill learning to be applied to larger continuous domains (Mohamed & Rezende, 2015; Karl et al., 2017; Gregor et al., 2016; Eysenbach et al., 2018; Sharma et al., 2019; Li et al., 2019; Hansen et al., 2020). However, these methods were limited in the size of skillsets they were able to learn as they only maximize a loose lower bound on mutual information, making it difficult to accurately measure the diversity of a skillset (Levy et al., 2024).

Related to empowerment-based skill learning is unsupervised goal-conditioned reinforcement learning (GCRL) that learn goal-conditioned skills using some automated curriculum that expands the distribution over goal states over time (Ecoffet et al., 2019; Mendonca et al., 2021; Nair et al., 2018; Pong et al., 2019; Campos et al., 2020; Pitis et al., 2020; Held et al., 2017; McClinton et al., 2021). The problem with GCRL is that in significantly stochastic settings where specific states cannot be consistently achieved, the GCRL objective also becomes a loose lower bound on the mutual information between skills and states, providing an agent with a weak signal for learning large skillsets. In contrast, Skillset Empowerment and our approach learn tighter bounds on mutual information, providing a dense signal for how to learn increasingly diverse skillsets.

Also, related to our work is the large body of research for building world models in order to learn new representations (Hafner et al., 2019; Ha & Schmidhuber, 2018; Gregor et al., 2019; Grill et al., 2020; Guo et al., 2022; Ghugare et al., 2023; Assran et al., 2023; Bardes et al., 2024; Pathak et al., 2017). Learning a world model is challenging because the full state space needs to be encoded into a single compressed latent space in order to learn a new state representation. In contrast, LPE does not learn models to learn a new state representation but rather to determine how many distinct actions are available in a state. To do this, LPE groups together redundant actions that achieves the same terminating states, which only requires encoding the more limited set of states $s_n$ that are reachable in a small number $n$ actions.

## 6 CONCLUSION

Empowerment has the potential to help agents become general purpose agents with large skillsets, but this potential may never be realized as long as measuring the empowerment of a state requires a simulator of the environment. In this work, we takes a step toward a more scalable way to compute empowerment by presenting a method that can measure empowerment using only a latent-predictive model. We show empirically in a variety of settings that our approach can learn equally-sized skillsets as the leading empowerment algorithm that requires access to a simulator of the environment.

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

## A  SKILLSET EMPOWERMENT ACTOR-CRITIC ARCHITECTURE

This section reviews how Skillset Empowerment maximizes the variational mutual information objective with respect to the skillset distributions $\phi$ and $\pi$ as we will use a similar optimization architecture in our approach. In order to optimize the variational mutual information $\tilde{I}(Z; S_n|s_0, \phi, \pi)$ with respect to $\phi$ and $\pi$ using deep learning, Skillset Empowerment first vectorizes these distributions. Skillset Empowerment represents the distribution over skills $\phi$ with a scalar representing the side length of a uniform distribution in the shape of a $d$-dimensional cube. For instance, if the skill space has two dimensions (i.e., $d = 2$), skills are uniformly sampled from a square centered at the origin with side length $\phi$. Figure 3 provides an illustration of this distribution over skills. Skillset Empowerment represents the skill-conditioned policy $\pi$ as a vector, which contains the weights and biases of the neural network $f_\pi : \mathcal{S} \times \mathcal{Z} \to \mathcal{A}$ that when given a skill start state $s_0$ and skill $z$, outputs the mean of a diagonal gaussian skill-conditioned policy $\pi(a|s_0, z)$ with a fixed standard deviation. That is,

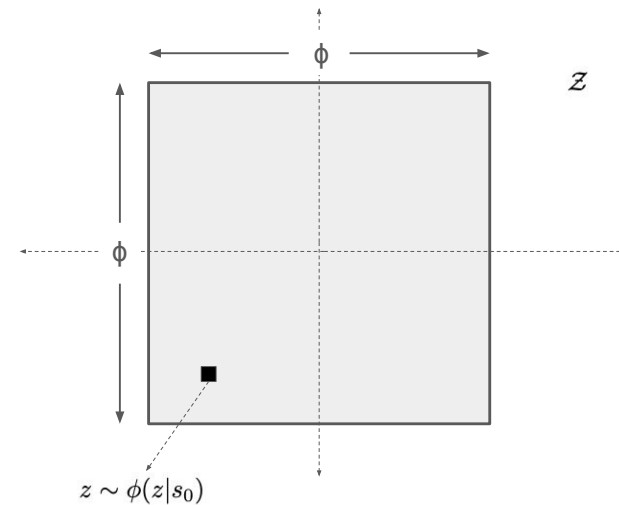

Figure 3: Illustration of the uniform distribution over skills $\phi$ used by Skillset Empowerment and our approach. The uniform distribution takes the shape of a $d$-dimensional cube centered at the origin with side length $\phi$. For instance, if the dimensionality of the skill space is 2 (i.e., $d = 2$) as in the figure, skills $z \sim \phi(z|s_0)$ are uniformly sampled from a square centered at the origin with side length $\phi$.

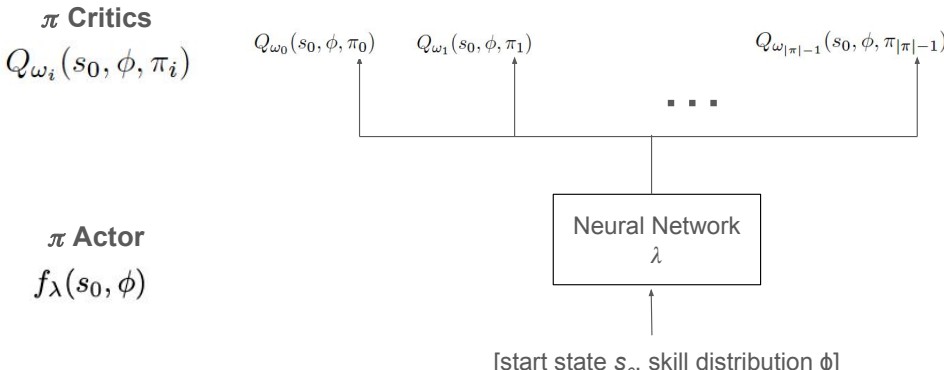

Figure 4: Illustration of how the parameter-specific critics, $Q_{\omega_i}$ for $i = 0 \ldots, |\pi| - 1$, attach to the actor $f_\lambda$ in order to determine the gradients of the actor. For each parameter $i$ in $\pi$, a critic $Q_{\omega_i}$ approximates how the diversity of the skill-conditioned policy changes with small changes to the $i$-th parameter of $\pi$. To obtain gradients showing how the diversity of a skill-conditioned policy changes with respect to $\lambda$, gradients are thus passed through each of the parameter-specific critics.

the skill-conditioned policy distribution $p(a|s_0, z, \pi) = \pi(a|s_0, z) = \mathcal{N}(a; \mu = f_\pi(s_0, z), \sigma = \sigma_0)$, in which the standard deviation $\sigma_0$ is a hyperparameter set by the user.

Using these vectorized forms of $\phi$ and $\pi$, Skillset Empowerment maximizes the variational mutual information objective using two actor-critic structures that are nested. The purpose of the inner actor-critic is to learn a policy (i.e., actor) $f_\lambda : \mathcal{S} \times \phi \to \pi$ that takes as input the skill start state $s_0$ and a scalar value $\phi$ representing the shape of the distribution over skills and outputs the vector $\pi$ representing a diverse skill-conditioned policy. To guide the actor to more diverse skill-conditioned policy in a tractable manner, Skillset Empowerment trains a critic for each of the $|\pi|$ parameters in the $\pi$ vector. The critic for the $i$-th parameter, $Q_{\omega_i} : \mathcal{S} \times \phi \times \pi \to \mathbb{R}$ will take as input a skill start state $s_0$, a skill distribution parameter $\phi$, and the $i$-th parameter of the skill-conditioned policy $\pi$. This scalar is used to represent a skill-conditioned policy equal to greedy output of the actor $f_\lambda(s_0, \phi)$, except for the $i$-th parameter which can take on noisy values. The critic $Q_{\omega_i}$ is trained to output an

approximation of the mutual information of skillsets defined by $\phi$ and $\pi$, which can contain noisy values for the $i$-th parameter. See Figure 4 for a visualization of how the $|\pi|$ critics attach to the $f_\lambda$ actor to determine the gradients with respect to the parameters $\lambda$ of the actor. The purpose of the outer actor-critic, is to train the policy $f_\mu : \mathcal{S} \to \phi$, which takes as input a skill start state $s_0$ and outputs a scalar value representing a distribution over skills. To guide this policy to outputting more diverse skillsets, a critic is learned to approximate the variational mutual information for various skillsets defined by $(s_0, \phi, \pi = f_\lambda(s_0, \phi))$.

## B    Visualization of diversity score distributions

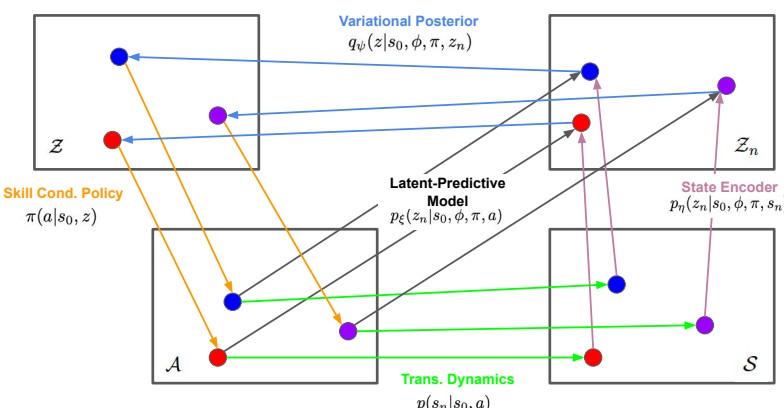

Figure 5: Illustration of trained latent-predictive, state encoding, and variational posterior distributions for a diverse skillset. Per the image, the latent-predictive models (black arrows) output $z_n$ that (i) match the output of the state encoding distribution (pink arrows) and (ii) are unique and can be decoded back to the original skill via the variational posterior (blue arrows).

## C    How LPE Explores the Space of Skill-Conditioned Policies

Even though LPE agents only interact with the environment by greedily following its nearly deterministic skill-conditioned policy, equation 6 has built-in mechanisms that encourage agents to try different skillsets that execute new actions. When LPE measures the skillset diversity of a large number of candidate skillsets that contain small changes to one of the parameters of $\pi$, there will be skillset candidates $(\phi, \pi)$ that incorporate new actions into the skillset that are not in the replay buffer as they have not been executed by previous skillsets. For these skillsets that execute new actions in addition to the previously discovered unique actions, the $\tilde{I}(Z; Z_n|s_0, \phi, \pi)$ term, which measures the number of actions in a skillset, may increase which then pushes up the overall diversity score for these candidate skillsets. The higher diversity score will in turn encourage the agent to "explore" by changing its skillset to include skills that execute these new actions. If the agent does update its skillset, once the new actions are executed in the environment and the states $s_n$ have been observed, the agent can keep this action in its skillset (i.e., continue to have some skill $z$ execute this action) if it targets some new state $s_n$ or remove the action from the skillset if $s_n$ can already be achieved by some other skill in the skillset.

## D    Proof of $J(s_0, \phi, \pi)$ as a lower bound to $I_J(s_0, \phi, \pi)$

Below we prove that the skillset diversity objective $J(s_0, \phi, \pi)$ used in equation 6 is a lower bound to the $I_J$ objective in equation 7. For this proof, let the joint distributions $p_0(z, a, s_n, z_n)$ and $p_1(z, a, s_n, z_n)$ be defined as follows:

$$p_0(z, a, s_n, z_n|s_0, \phi, \pi) = \phi(z|s_0)\pi(a|s_0, z)p(s_n|s_0, a)p_\eta(z_n|s_0, \phi, \pi, s_n)$$
$$p_1(z, a, s_n, z_n|s_0, \phi, \pi) = \phi(z|s_0)\pi(a|s_0, z)p(s_n|s_0, a)p_\xi(z_n|s_0, \phi, \pi, a)$$

The difference between the two joint distribution is that $p_0$ generates $z_n$ using the state encoding distribution $p_\eta$, while $p_1$ generates $z_n$ using the latent-predictive model $p_\xi$. Then

$$I_J(s_0, \phi, \pi) = H(Z|s_0, \phi) + \log(\mathbb{E}_{(z,a,s_n,z_n)\sim p_0}[p(z|s_0, \phi, \pi, z_n)])$$

$$= H(Z|s_0, \phi) + \log\left(\mathbb{E}_{(z,a,s_n,z_n)\sim p_1}\left[\frac{p_0(z, a, s_n, z_n)}{p_1(z, a, s_n, z_n)} p(z|s_0, \phi, \pi, z_n)\right]\right) \tag{8}$$

$$\geq H(Z|s_0, \phi) + \mathbb{E}_{(z,a,z_n)\sim p_1}[\log p(z|s_0, \phi, \pi, z_n)] \tag{9}$$

$$- \mathbb{E}_{(a,s_n)\sim p_1}[D_{KL}(p_\xi(z_n|s_0, \phi, \pi, a)||p_\eta(z_n|s_0, \phi, \pi, s_n))]$$

$$= I(Z; Z_n|s_0, \phi, \pi) - \mathbb{E}_{(a,s_n)\sim p_1}[D_{KL}(p_\xi(z_n|s_0, \phi, \pi, a)||p_\eta(z_n|s_0, \phi, \pi, s_n))]$$

$$\geq \tilde{I}(Z; Z_n|s_0, \phi, \pi) - \mathbb{E}_{(a,s_n)\sim p_1}[D_{KL}(p_\xi(z_n|s_0, \phi, \pi, a)||p_\eta(z_n|s_0, \phi, \pi, s_n))] \tag{10}$$

In line 8, importance sampling is used to integrate the latent-predictive model (found in the joint distribution $p_1$) into the objective. The inequality in line 9 is due to Jensen's Inequality. The KL divergence term results because all distributions in the $p_0/p_1$ ratio cancel out except for the state encoding distribution $p_\eta(z_n|s_0, \phi, \pi, s_n)$ and the latent-predictive model $p_\xi(z_n|s_0, \phi, \pi, a)$. The last inequality 10 results from replacing the true mutual information $I(Z; Z_n|s_0, \phi, \pi)$ with the variational mutual information $\tilde{I}(Z; Z_n|s_0, \phi, \pi)$ (Barber & Agakov, 2003).

## E  PROOF THAT $I(Z; S_n|s_0, \phi, \pi)$ AND $J(s_0, \phi, \pi)$ HAVE THE SAME OPTIMAL $\pi$ UNDER CERTAIN ASSUMPTIONS

**Assumption 1:** There exists a finite maximum posterior $p_{\max}$ for the following posteriors: $p(z|s_0, \phi, \pi, s_n)$, $p(z|s_0, \phi, \pi, z_n)$, and $q_\phi(z|s_0, \phi, \pi, z_n)$.

**Assumption 2:** There exists one or more (skill start state $s_0$, skill distribution $\phi$) tuples such that there is also a skill-conditioned policy $\pi^*$ in which the variational posterior $q_\phi(z|s_0, \phi, \pi^*, z_n) = p_{\max}$ for all $(z, z_n)$ with nonzero probability and the KL divergence $D_{KL}(p_\xi||p_\eta) = 0$ for all $(a, s_n)$ tuples with non-zero probability.

Given the two assumptions above, note that for a certain skill distribution size $\phi$, the following quantities: (a) $I(Z; S_n|s_0, \phi, \pi)$, (b) $I(Z; Z_n|s_0, \phi, \pi)$, in which $z_n$ is sampled from the state encoding distribution $p_\eta$, (c) $I_J(s_0, \phi, \pi)$ in equation 7, and (d) our diversity objective $J(s_0, \phi, \pi)$ in equation 6, cannot be larger than $H(Z|s_0, \phi) + \log p_{\max}$ and the maximum occurs when the posterior in each term equals $p_{\max}$ because, as a result of assumption 1, the expectation of posteriors cannot be larger than $p_{\max}$ and $H(Z|s_0, \phi)$ is a constant for a given $\phi$.

For a $(s_0, \phi)$ tuple from Assumption 2, $\pi^*$ is an optimal skillset for the $J(s_0, \phi, \pi)$ objective because $J(s_0, \phi, \pi^*) = H(Z|s_0, \phi) + \log p_{\max}$. $\pi^*$ also maximizes the upper bound of $J(s_0, \phi, \pi)$, $I_J(s_0, \phi, \pi)$, as $I_J(s_0, \phi, \pi^*)$ must equal $H(Z|s_0, \phi) + \log p_{\max}$ because it is both at least as large as $H(Z|s_0, \phi) + \log p_{\max}$ because it upper bounds $J(s_0, \phi, \pi^*)$ and also less than or equal to $H(Z|s_0, \phi) + \log p_{\max}$ because it cannot take on a higher value as noted in the prior paragraph. Given that $I_J$ is at its maximum, then for the skillset $(s_0, \phi, \pi^*)$, the posterior $p(z|s_0, \phi, \pi, z_n) = p_{\max}$ for $(z, z_n)$ with non-zero probability. This in turn means that $I(Z; Z_n|s_0, \phi, \pi)$, in which $z_n$ is sampled from the state encoding distribution, equals $I_J(S_0, \phi, \pi)$ because the log expectation of a constant equals the expected log of a constant. Finally, the mutual information between skills and states $I(Z; S_n|s_0, \phi, \pi^*)$ also must equal $H(Z|s_0, \phi) + \log p_{\max}$ because it is at least as large as $I(Z; Z_n|s_0, \phi, \pi)$ due to the data processing inequality but is also at most $H(Z|s_0, \phi) + \log p_{\max}$ because that is its maximum value. Thus, for the one or more tuples $(s_0, \phi)$ from Assumption 2, $\pi^*$ maximizes both $J(s_0, \phi, \pi)$ and $I(Z; S_n|s_0, \phi, \pi)$ objectives.

**Commentary:** The assumptions listed above are reasonable. The first assumption is realistic because if the skill-conditioned policy $\pi(a|s_0, z)$ has some stochasticity and only models continuous functions, then there is a limit to how tightly the distinct skills can be "packed" into the skillset. That is, for some small change in the skill $z$, there will realistically be some overlap in the states $s_n$ that are targeted, which puts a limit on the tightness of the posterior distribution. Our results also show that the second assumption is realistic as our agents are able to learn skillsets with tight posterior distributions (i.e., high $q_\psi(z|s_0, \phi, \pi, z_n)$) and accurate latent-predictive models (i.e., low $D_{KL}(p_\xi||p_\eta)$).

## F    SKILL-CONDITIONED POLICY ACTOR-CRITIC OBJECTIVE FUNCTIONS

The process of training the parameter-specific critics $Q_{\omega_0}, \ldots, Q_{\omega_{|\pi|-1}}$ in parallel follows a three step process. Note that to approximate the parameter-specific critics, we will use parameter-specific latent-predictive models $p_{\xi_i}$, state encoding distributions $p_{\eta_i}$, and variational posterior distributions $q_{\psi_i}$ for $i = 0, \ldots, |\pi| - 1$.

In the first step, the diversity scores for various noisy $(\phi, \pi_i)$ skillsets are maximized by maximizing the following objective with respect to the latent-predictive model $p_{\xi_i}$, the state encoder distribution $p_{\eta_i}$, and the variational posterior $q_{\psi_i}$ for all parameters $i = 0, \ldots, |\pi| - 1$ in parallel:

$$J_i(\xi_i, \eta_i, \psi_i) = \mathbb{E}_{s_0 \sim \beta, \phi \sim \hat{f}_\mu, \pi_i \sim \hat{f}_\lambda}[\tilde{I}(Z; Z_n|s_0, \phi, \pi_i)] \tag{11}$$
$$+ \mathbb{E}_{(a, s_n|s_0) \sim \beta}[D_{KL}(p_{\xi_i}(z_n|s_0, \phi, \pi_i, a)||p_{\eta_i}(z_n|s_0, \phi, \pi_i, s_n))]]$$
$$= \mathbb{E}_{s_0 \sim \beta, \phi \sim \hat{f}_\mu, \pi_i \sim \hat{f}_\lambda}[\mathbb{E}_{a \sim \pi(a|s_0, z), z_n \sim p_{\xi_i}(z_n|s_0, \phi, \pi, a)}[\log q_{\psi_i}(z|s_0, \phi, \pi, z_n)]$$
$$+ \mathbb{E}_{(a, s_n|s_0) \sim \beta}[D_{KL}(p_{\xi_i}(z_n|s_0, \phi, \pi_i, a)||p_{\eta_i}(z_n|s_0, \phi, \pi_i, s_n))]]$$

For the $i$-th critic, the outer expectation sampling $(s_0, \phi, \pi)$ will sample $s_0$ from the replay buffer $\beta$, $\phi$ by adding noise to the greedy value of $\phi = f_\mu(s_0)$, and the scalar $\pi_i$ by adding noise to the $i$-th parameter of the skill-conditioned policy $\pi = f_\lambda(s_0, \phi)$. Note that this is the same diversity-measuring objective as $J(s_0, \phi, \pi)$ in equation 6 except the (action $a$, skill-terminating state $s_n$) tuples are sampled from a replay buffer $\beta$ containing $(s_0, a, s_n)$ transitions. In addition, because the latent-predictive model $p_{\xi_i}$ is implemented as a diagonal gaussian distribution, the reparameterization trick (Kingma & Welling, 2022) can be used to simplify the gradient through the $p_{\xi_i}$ distribution which appears both in the $\tilde{I}(Z; Z_n|s_0, \phi, \pi_i)$ and the KL divergence terms.

In the second step, we approximate the KL divergence between the latent-predictive model $p_{\xi_i}$ and the state encoder $p_{\eta_i}$ for various $(s_0, \phi, \pi_i, a, s_n)$ combinations. This will be needed in order to accurately compute the diversity score for a particular $(\phi, \pi_i)$ skillset without needing a simulator to sample the skill-terminating state $s_n$. To approximate the KL divergence, we minimize the following objective with respect to $\kappa_i$ for all $i = 0, \ldots, |\pi| - 1$ in parallel, in which $\kappa_i$ represent the parameters of the neural network that approximates the KL divergence.

$$J_i(\kappa_i) = \mathbb{E}_{s_0 \sim \beta, \phi \sim \hat{f}_\mu, \pi_i \sim \hat{f}_\lambda, (a, s_n|s_0) \sim \beta}[(Q_{\kappa_i}(s_0, \phi, \pi_i, a) - \text{Target}(s_0, \phi, \pi_i, a, s_n))^2], \tag{12}$$
$$\text{Target}(s_0, \phi, \pi_i, a, s_n) = \mathbb{E}_{z_n \sim p_{\xi_i}(z_n|s_0, \phi, \pi_i, a)}[\log p_{\xi_i}(z_n|s_0, \phi, \pi_i, a) - \log p_{\eta_i}(z_n|s_0, \phi, \pi_i, s_n)]$$

In the third step, the parameter-specific critics $Q_{\omega_0}, \ldots, Q_{\omega_{|\pi|-1}}$ are trained to approximate the $J(s_0, \phi, \pi)$ diversity score using the updated parameter-specific latent-predictive model $p_{\xi_i}$, variational posterior $q_{\psi_i}$, and KL approximation parameters $\kappa_i$. This is done by minimizing the following supervised learning objective with respect to the parameters $\omega_i$.

$$J_i(\omega_i) = \mathbb{E}_{s_0 \sim \beta, \phi \sim \hat{f}_\mu, \pi_i \sim \hat{f}_\lambda}[(Q_{\omega_i}(s_0, \phi, \pi_i) - \text{Target}(s_0, \phi, \pi_i))^2], \tag{13}$$
$$\text{Target}(s_0, \phi, \pi_i) = \mathbb{E}_{a \sim \pi_i(a|s_0, z), z_n \sim p_{\xi_i}(z_n|s_0, \phi, \pi_i, a)}[\log q_{\psi_i}(z|s_0, \phi, \pi_i, z_n) - Q_{\kappa_i}(s_0, \phi, \pi_i, a)]$$

The skill-conditioned policy actor $f_\lambda$ is then trained to output more diverse skill-conditioned policies by maximizing the following objective with respect to the parameters $\lambda$:

$$J(\lambda) = \mathbb{E}_{s_0 \sim \beta, \phi \sim \hat{f}_\mu}\left[\sum_{i=1}^{|\pi|-1} Q_{\kappa_i}(s_0, \phi, f_\lambda(s_0, \phi)[i])\right], \tag{14}$$

in which $f_\lambda(s_0, \phi)[i]$ outputs the $i$-th parameter in $\pi$. Figure 4 provides a visualization of how the parameter-specific critics $Q_{\kappa_i}$ are attached the actor $f_\lambda$ in order to determine the gradients of the $J(s_0, \phi, \pi)$ diversity score with respect to the parameters $\lambda$ of the actor.

## G    VAE OBJECTIVE FOR TRAINING LATENT-PREDICTIVE MODEL

To train the latent-predictive model $p_\xi(z_n|s_0, \phi, a)$ to match the data distribution for various values of $\phi$ we will use a VAE generative model. Given that $p_\xi$ is modeled using a VAE, $p_\xi(z_n|s_0, \phi, a)$ is

a marginal of the joint distribution $p_\xi(c, z_n|s_0, \phi, a) = p_{\xi_c}(c|s_0, \phi, a)p_{\xi_d}(z_n|s_0, \phi, a, c)$, in which $p_{\xi_c}(c|s_0, \phi, a)$ is the prior distribution of the VAE that outputs a latent code $c$. $p_{\xi_d}(z_n|s_0, \phi, a, c)$ is the decoder of the VAE that outputs a $z_n$ given $s_0$, $\phi$, $a$, and latent code $c$. The VAE will also make use of a variational posterior distribution $q_{\xi_v}(c|s_0, \phi, a, z_n)$ which outputs a distribution over the latent code $c$ given a $z_n$. The data distribution that the latent-predictive model is trying to match is $p_D(z_n|s_0, \phi, a)$, which is the marginal of the joint distribution $p_\eta(s_n, z_n|s_0, \phi, a) = p(s_n|s_0, a)p_\eta(z_n|s_0, \phi, \pi = f_\lambda(s_0, \phi), s_n)$. $p_\eta(z_n|s_0, \phi, \pi = f_\lambda(s_0, \phi), s_n)$ is the state-encoding distribution learned when optimizing the skill-conditioned policy objective in equation 6.

The following objective is minimized with respect to the VAE parameters $(\xi_p, \xi_d, \xi_v)$ to train the latent-predictive model to match the data distribution:

$$J_{\text{VAE}}(\xi_p, \xi_d, \xi_v) = \mathbb{E}_{(s_0,a)\sim\beta, \phi\sim\hat{f}_\mu}[D_{KL}(p_D(z_n|s_0, \phi, a)||p_\xi(z_n|s_0, \phi, a)) + \mathbb{E}_{z_n\sim p_D(s_0,\phi,a)}[ \quad (15)$$
$$D_{KL}(q_{\xi_v}(c|s_0, \phi, a, z_n)||p_\xi(c|s_0, \phi, a, z_n))]]]$$
$$= \mathbb{E}_{(s_0,a)\sim\beta, \phi\sim\hat{f}_\mu, z_n\sim p_D(z_n|s_0,\phi,a), c\sim q_{\xi_v}(c|s_0,\phi,a,z_n)}[\log q_{\xi_v}(c|s_0, \phi, a, z_n)$$
$$- \log p_{\xi_p}(c|s_0, \phi, a) - \log p_{\xi_d}(z_n|s_0, \phi, a, c)]$$

# H  SKILL DISTRIBUTION ACTOR-CRITIC OBJECTIVE FUNCTIONS

The critic functions $Q_\rho(s_0, \phi)$ are trained using a two step procedure. In the first step, for a variety of noisy $\phi$ values, the variational posterior parameters $\psi$ are updated so that a tighter bound between the variational mutual information $\tilde{I}(Z; Z_n|s_0, \phi)$ and the true mutual information $I(Z; Z|s_0, \phi)$ is achieved. This is done by maximizing the following maximum likelihood objective with respect to $\psi$.

$$J(\psi) = \mathbb{E}_{s_0\sim\beta, \phi\sim\hat{f}_\mu, z\sim\phi(z|s_0), z_n\sim p_\xi(z_n|s_0,\phi,z)}[\log q_\psi(z|s_0, \phi, z_n)], \quad (16)$$

in which the distribution $p_\xi(z_n|s_0, \phi, z)$ is the marginal of the joint distribution $p(\pi, a, z_n|s_0, \phi, z) = \pi(a|s_0, z)p_\xi(z_n|s_0, \phi, a)$ if $\pi = f_\lambda(s_0, \phi)$ and 0 otherwise. $p_\xi(z_n|s_0, \phi, a)$ is the latent-predictive model learned by the VAE generative model. Note that set of variational posterior parameters $\psi$ used in the actor-critic update for $\phi$ is different than the set of variational parameters used in the actor-critic update for the skill-conditioned policy.

In the second step, the critic $Q_\rho(s_0, \phi)$ is trained to approximate the diversity score of the $(\phi, \pi = f_\lambda(s_0, \phi))$ skillset using the updated variational posterior $\psi$ parameters. This is done by minimizing the following supervised learning objective with respect to $\rho$:

$$J(\rho) = \mathbb{E}_{s_0\sim\beta, \phi\sim\hat{f}_\mu}[(Q_\rho(s_0, \phi) - \text{Target}(s_0, \phi))^2], \quad (17)$$
$$\text{Target}(s_0, \phi) = \mathbb{E}_{z\sim\phi(z|s_0), z_n\sim p_\xi(z_n|s_0,\phi,z)}[\log q_\psi(z|s_0, \phi, z_n)].$$

The actor is then updated my maximizing the following objective with respect to $\mu$:

$$J(\mu) = \mathbb{E}_{s_0\sim\beta}[Q_\rho(s_0, f_\mu(s_0))] \quad (18)$$

# I  GPU INFORMATION

All experiments were done with either 4 H100 SXM (80GB VRAM/GPU) or 8 RTX 4090 GPUs (24GB VRAM/GPU) rented from RunPod. The continuous mountain car domain required 1-2 hours of training. The stochastic four rooms and RGB QR code domains required 1-4 hours of training.

# J  ENVIRONMENT DETAILS

1. Stochastic Four Rooms Navigation
   - State dim: 2
   - Action space: Continuous
   - Action Dim: 2

- Action range per dimension: $[-1, 1]$ reflecting $(\Delta x, \Delta y)$ for position of agent
- $p(s_0)$ is a single $(x, y)$ position
- $n = 5$ primitive actions

2. Stochastic Four Rooms Pick-and-Place

- State dim: 4
- Action space: Continuous
- Action Dim: 4
- Action range per dimension: $[-1, 1]$. First two dimensions reflect $(\Delta x, \Delta y)$ change in position for agent and the second two dimensions reflect the change in position for the object. The object can only be moved by the amount specified in the final two dimensions of the action if the object is within two units.
- $p(s_0)$ is a single $(x_{\text{agent}}, y_{\text{agent}}, x_{\text{object}}, y_{\text{object}})$ start state
- $n = 5$ primitive actions

3. RGB QR Code Navigation

- State dim: 2
- Action Dim: 2
- Action space: Discrete
- Action Range: $[-1, 1]$. First dimension reflects the horizontal movement. If first dimension is in range $\in [-1, -\frac{1}{3}]$, agent moves left. If first dimension is in range $[\frac{1}{3}, 1]$, agent moves right. Otherwise the agent does not make a horizontal movement. The second dimension reflects the north-south movement following the same pattern.
- The RGB color vector for the colored squares in the QR code background is a 3-dim vector, in which each component is randomly sampled from the range $[0.7, 1]$ (i.e., has a light color). The agent is shown with a 2x2 set of black squares.
- $p(s_0)$ is a single start state in the center of the room with a white background
- $n = 5$ primitive actions

4. RGB QR Code Pick-and-Place

- State dim: 4
- Action Dim: 4
- Action space: Discrete
- Action Range: $[-1, 1]$. First two dimensions are same as navigation task. The second two reflect how the object will be moved provided the object is within two units.
- The RGB color vector for the colored squares in the QR code background is a 3-dim vector, in which each component is randomly sampled from the range $[0.7, 1]$ (i.e., has a light color). The agent is shown with a 2x2 set of black squares. The object is shown with a 2x2 set of yellow squares.
- $p(s_0)$ is a single start state in which the agent and object are in same position in the center of the room with a white background
- $n = 5$ primitive actions

5. Continuous Mountain Car

- State dim: 2
- Action space: Continuous
- Action Dim: 1
- Action range per dimension: $[-1, 1]$
- $p(s_0)$ is a single x position and velocity.
- $n = 10$ primitive actions

## K  IMPLEMENTING HIERARCHICAL AGENTS WITH LPE

Coding hierarchical agents that use the $(\phi, \pi)$ LPE skillsets as a temporally extended action space is simple. For the higher level policy $\pi : \mathcal{S} \to \mathcal{Z}$ that outputs a skill $z$ from the LPE skillset given some

state, attach a tanh activation function to this policy, which bounds the output to $[-1, 1]$, and then multiply that output by $\phi$, which will bound the skill action space to $[-\phi, \phi]$ in every dimension, which has the same shape as $d$-dimensional cubic distribution that $\phi$ represents. (Note that in our implementation, $\phi$ is technically the log of the half length of each side of the $d$-dimensional cubic uniform distribution so the output of the tanh activation function should be multiplied by $e^\phi$. We have $\phi$ represent the log of the half length of side so the $\phi$ actor $f_\mu$ can output negative numbers.) Then once a skill $z$ has been sampled, the skill can be passed to the LPE skill-conditioned policy $\pi(a|s_0, z)$ which will then output an action sequence that can then be executed in the environment.

## L  MUTUAL INFORMATION ENTROPY VISUALIZATIONS

Please refer to Figures 6-17 for visuals of the $H(S_n)$, $H(S_n|Z)$, $H(Z)$, $H(Z|S_n)$ mutual information entropy terms both before and after training.

**Stochastic Four Rooms Navigation – Post-Training**

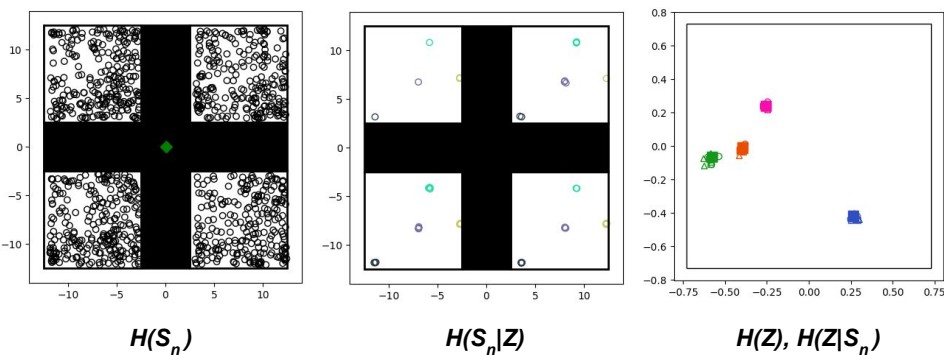

Figure 6: Entropy visualizations for a **trained** LPE agent in the stochastic four rooms navigation task. The $H(S_n)$ visual (left image) shows the skill-terminating states $s_n$ (i.e., the ending $(x, y)$ agent location) generated by 1000 skills randomly sampled from the learned $(\phi, \pi)$ skillset. Per the image, the skillset nearly uniformly targets the reachable state space. The $H(S_n|Z)$ visual shows the $s_n$ targeted by four randomly selected skills $z$ from the skillset, and each color shows the $s_n$ belonging to a different skill. For instance, the gold-colored $s_n$ shows a skill that targets the right side of a room. In the $H(Z), H(Z|S_n)$ visual (right image), the inner black outlined square is the skill distribution $\phi$. The solid small colored squares are randomly sampled skills $z$, and the empty squares of the same colors are samples from the learned posterior $q_\psi(z|s_0, \phi, \pi, s_n)$, which tightly surround the executed skill $z$. Per all the images, the agent has learned a diverse $(\phi, \pi)$ skillset, in which different skills $z$ target different $s_n$.

**Stochastic Four Rooms Navigation – Pre-Training**

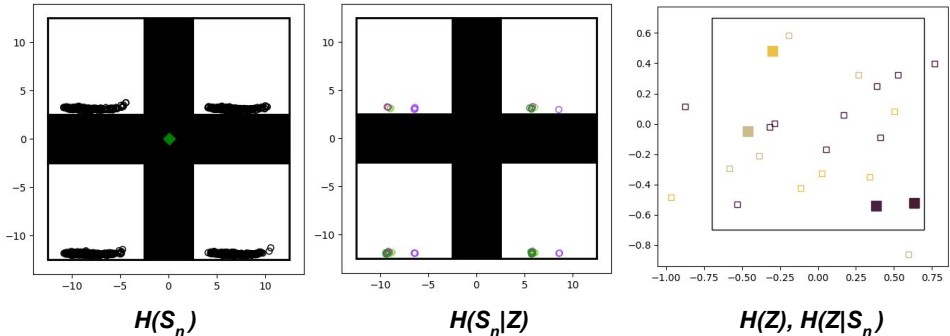

Figure 7: Entropy visualizations for a **non-trained** LPE agent in the stochastic four rooms navigation task. Per the poor state coverage in the left image and the high entropy posterior distributions in the right image, the agent does not start with a diverse skillset.

**Stochastic Four Rooms Pick-and-Place – Post-Training**

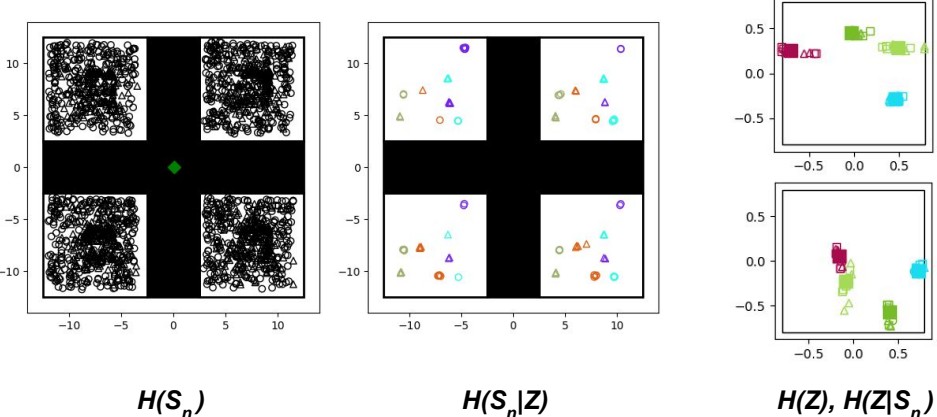

$H(S_n)$     $H(S_n|Z)$     $H(Z), H(Z|S_n)$

Figure 8: Entropy visualizations for a **trained** LPE agent in the stochastic four rooms pick-and-place task. In the $H(S_n)$ and $H(S_n|Z)$ visualizations, the agent location component of the skill-terminating state $s_n$ is marked by a circle, and the object location component is marked by a triangle. Per the center image, which shows the $s_n$ for four different skills $z$, each skill does some different behavior. The gold skill has the agent push the object towards the bottom left corner of any room. On the other hand, the purple skill consists mostly of the agent moving towards the top right corner of any room without carrying the object. Per the images, the agent has learned a large skillset, in which different skills target different locations for the agent and object.

**Stochastic Four Rooms Pick-and-Place – Pre-Training**

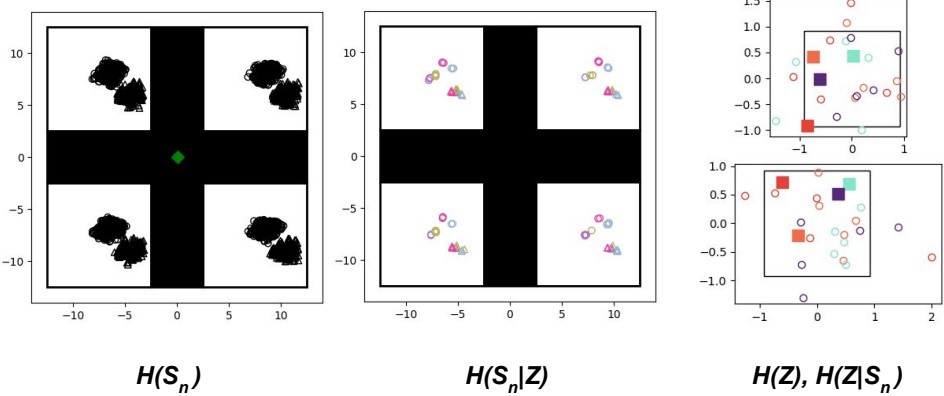

$H(S_n)$     $H(S_n|Z)$     $H(Z), H(Z|S_n)$

Figure 9: Entropy visualizations for a **non-trained** LPE agent in the stochastic four rooms pick-and-place task. Again, per the poor state coverage in the $H(S_n)$ visual and the high entropy posteriors in the $H(Z|S_n)$ visual, the agent does not start with a diverse skillset.

**RGB QR Code Navigation – Post-Training**

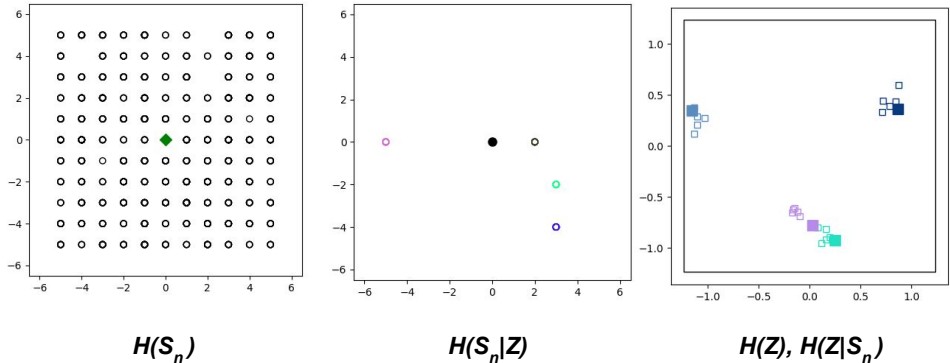

Figure 10: Entropy visualizations for a **trained** LPE agent in the RGB QR code navigation task. Note that $H(S_n)$ and $H(S_n|Z)$ plot the underlying state $s_n$ (i.e., the $(x, y)$ coordinate of the agent) that is not visible to the agent. In the RGB QR Code domains, the agent receives a 12x12x3 image (i.e., a 432-dim state). Per the images, the agent has learned a diverse skillset, in which different skills target different $(x, y)$ locations.

**RGB QR Code Navigation – Pre-Training**

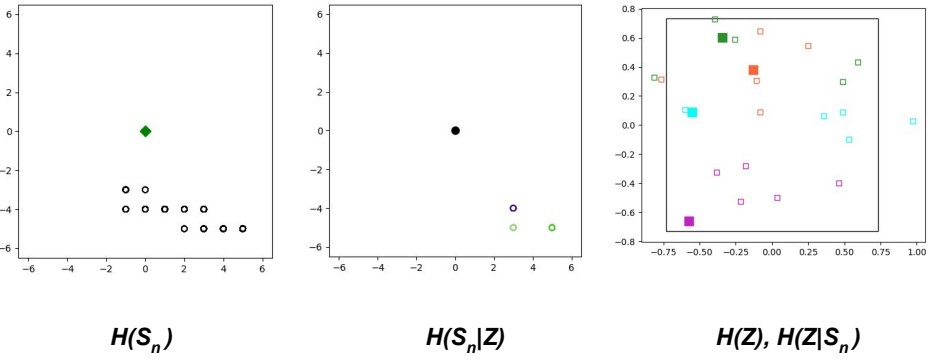

Figure 11: Entropy visualizations for a **non-trained** LPE agent in the RGB QR code navigation task. Per the visuals, the agent does not start with a diverse $(\phi, \pi)$ skillset.

**RGB QR Code Pick-and-Place – Post-Training**

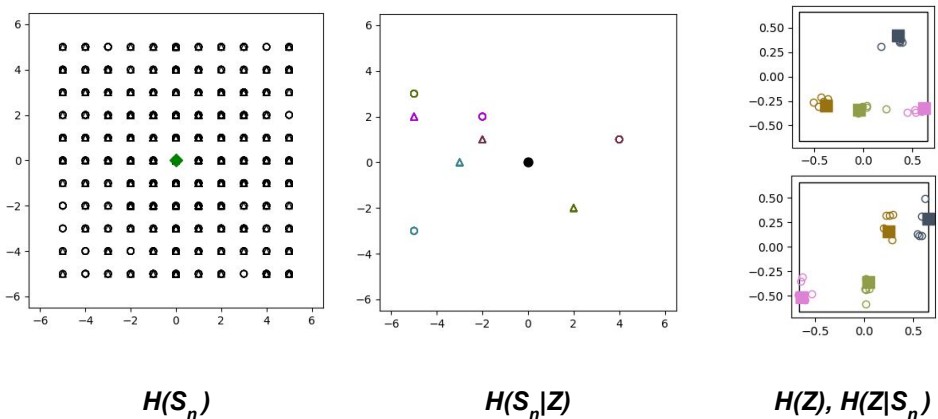

Figure 12: Entropy visualizations for a **trained** LPE agent in the RGB QR code pick-and-place task. Note that $H(S_n)$ and $H(S_n|Z)$ plot the underlying state $s_n$ (i.e., the $(x, y)$ coordinate of the agent) that is not visible to the agent. In the RGB QR Code domains, the agent receives a 12x12x3 image (i.e., a 432-dim state). In the visuals of the underlying state, the circles represent the agent location component of $s_n$ and the triangle represent the object location component of $s_n$. Per the images, the agent has learned a diverse skillset, in which different skills target different $(x, y)$ locations for the agent and object.

**RGB QR Code Pick-and-Place – Pre-Training**

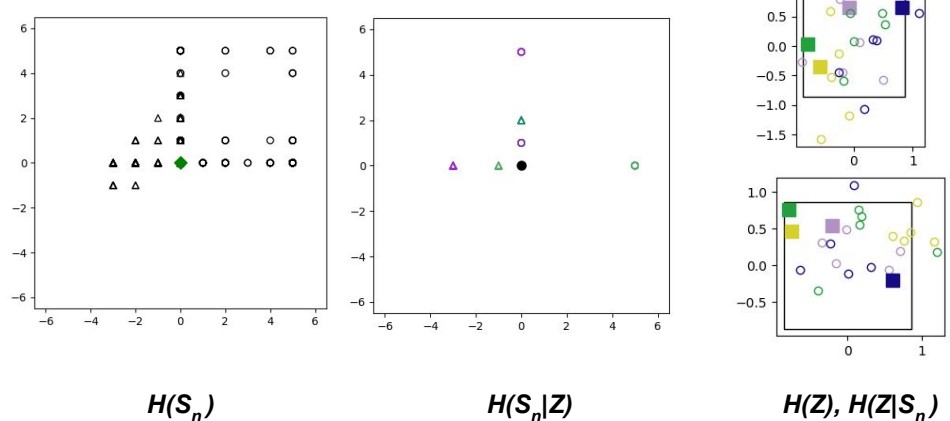

Figure 13: Entropy visualizations for a **non-trained** LPE agent in the RGB QR code pick-and-place task. Per the visuals, the agent does not start with a diverse $(\phi, \pi)$ skillset.

**Continuous Mountain car – Post-Training**

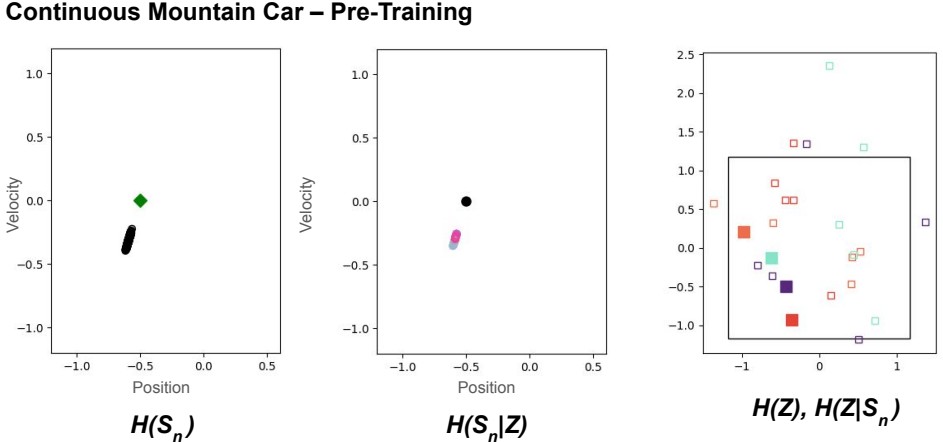

$H(S_n)$ $\quad$ $H(S_n|Z)$ $\quad$ $H(Z), H(Z|S_n)$

Figure 14: Entropy visualizations for a **trained** LPE agent in the continuous mountain car task. The x-axis in the $H(S_n)$ and $H(S_n|Z)$ visuals show the agent position component of $s_n$ and the y-axis shows the velocity component of $s_n$. The black dot in the $H(S_n|Z)$ shows the starting state for the mountain car agent. Per the images, the agent has learned a diverse skillset, in which skills target different tuples of (cart position, cart velocity).

**Continuous Mountain Car – Pre-Training**

$H(S_n)$ $\quad$ $H(S_n|Z)$ $\quad$ $H(Z), H(Z|S_n)$

Figure 15: Entropy visualizations for a **non-trained** LPE agent in the continuous mountain car task task. Per the visuals, the agent does not start with a diverse $(\phi, \pi)$ skillset.

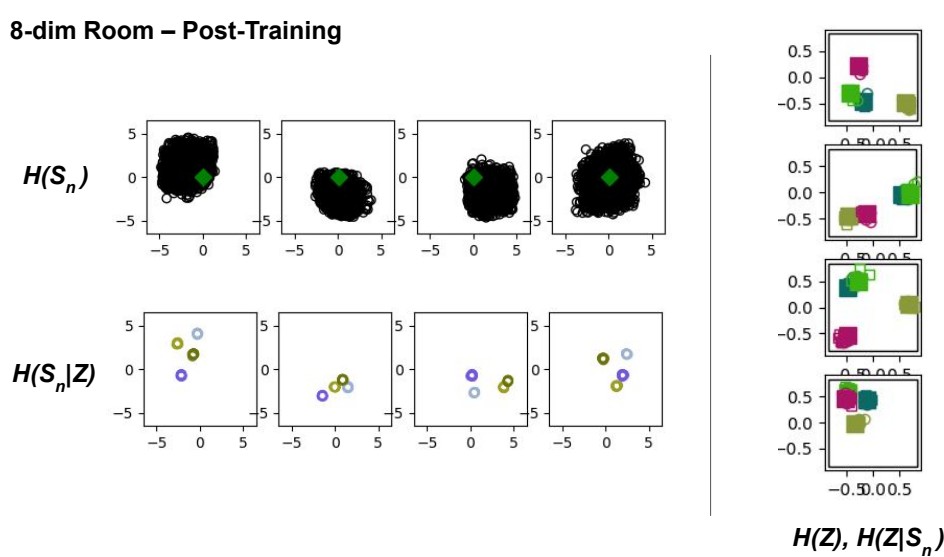

Figure 16: Entropy visualizations for a **trained** LPE agent in the eight dimension room task. Note that $H(S_n)$ and $H(S_n|Z)$ visuals have four plots in which each plots shows two dimensions of the skill-terminating state $s_n$. Per the visuals, the agent has learned a diverse $(\phi, \pi)$ skillset.

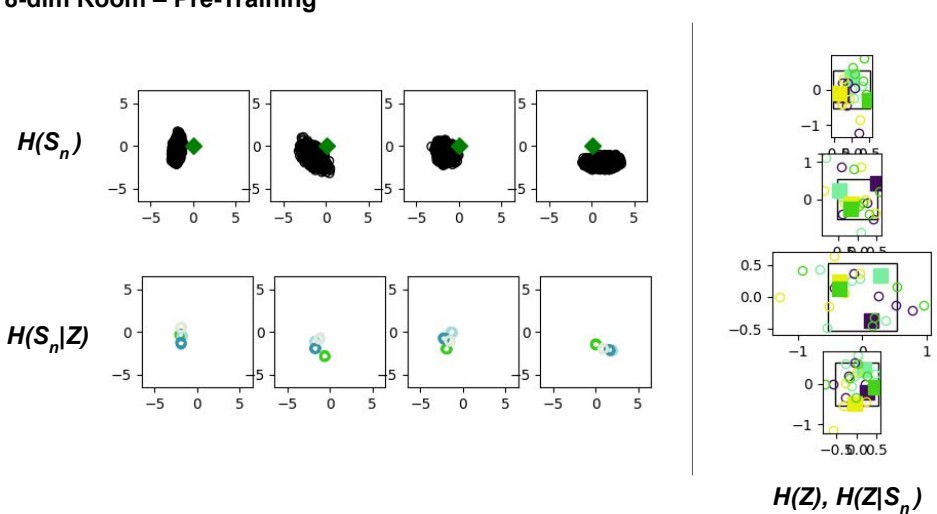

Figure 17: Entropy visualizations for a **non-trained** LPE agent in the eight dimension room task. Per the visuals, the agent does not start with a diverse $(\phi, \pi)$ skillset.

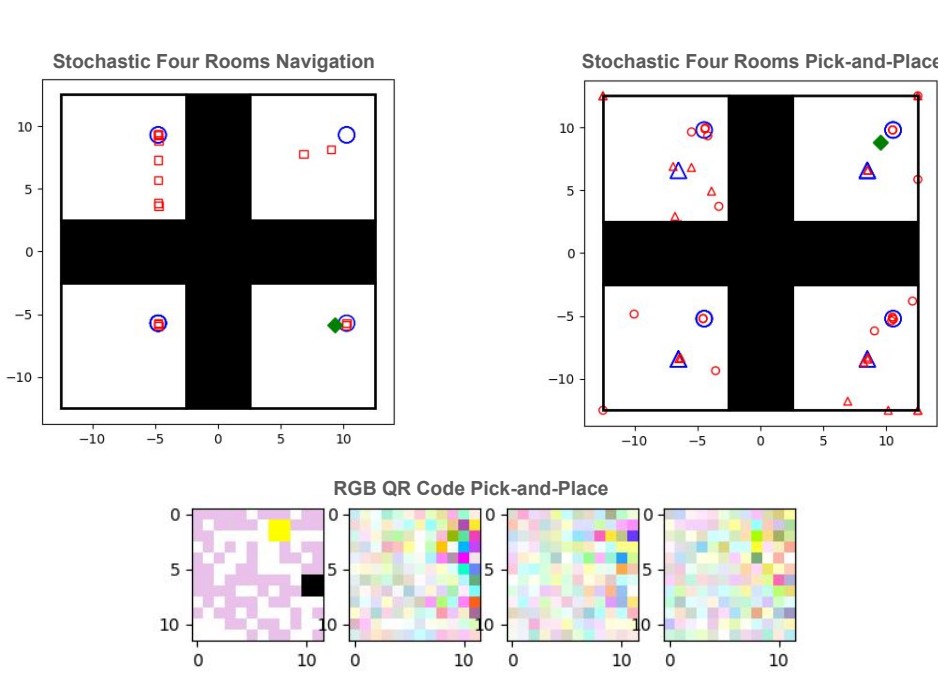

Figure 18: Examples of the challenges the VAE had in learning the transition dynamics in the stochastic domains. The top left image shows a result in the stochastic four rooms navigation domain. The blue circles show the correct next states (i.e., the next agent (x,y) location) when currently position at the green diamond. The red squares show 20 samples from the VAE model, in which 5 are significantly inaccurate. Note, that these are samples from a single step of the transition function. Over $n$ actions during an executed skill, there will be significantly more deviations from correct skill-terminating state. The top right image shows a sample from the pick-and-place version in which the VAE had even more difficulty. The blue triangle represents the correct next location for the object and the red triangles show samples of the object position from the VAE. The bottom image shows an example from the RGB QR Code pick-and-place task. The left image in the row shows the correct next observation. In this image, the black square is the agent, the yellow square is the object that can be manipulated, and the background is a pink QR code. The right three images show samples from the VAE, which provide a very inaccurate representation of the next state.

