# OpenReview forum: "Latent-Predictive Empowerment: Measuring Empowerment without a Simulator"
_ICLR.cc/2025/Conference — Submitted to ICLR 2025_

### Official Review · Reviewer_J8FD · 2024-10-19

**Soundness:** 2
**Presentation:** 2
**Contribution:** 2
**Rating:** 3
**Confidence:** 3

**Summary:**

The authors introduce a latent empowerment objective building upon a prior work, Skill Empowerment. They demonstrate that this latent empowerment objective learns more skills than the prior work.

**Strengths:**

**Method:** The objective is mathetmatically grounded and overall makes sense as a way to learn skills.

**Visualizations:** The authors do a good job of showing skill distributions in the appendix to help visualize the results.

**Weaknesses:**

**Novelty:** The method is an incremental improvement over Skill Empowerment (with the addition of latent skill prediction), evaluated on the exact same domains. Therefore the novelty with respect to the the experiments, insights, and method is pretty incremental.

**Clarity:** This paper is not well-written and is in many areas quite unclear. Some examples follow, but overall most subsections up until the experiments need rewriting:

- The introduction is not that clear; in the third paragraph, the description of LPE is too detailed with respect to talking about multual information terms and KL divergence terms. No equation is shown here (and it probably shouldn’t be), so the reader is forced to imagine the equation and think about why/how the statements the authors make are true. This would be much clearer if presented at a higher level.
    - Skills are also not defined in the introduction; skills can refer to many different definitions even in the space of reinforcement learning
    - Empowerment is also poorly defined in the 2nd sentence as: “the empowerment of a state measures the size of the largest skillset in that state”
    - Without knowing what skills mean in this context, empowerment is also hard to grasp for readers unfamiliar with prior works on empowerment. Can the authors put a better definition?
- Related to the above, Figure 1 is also not clear.
    - It’s hard to imagine a circle on a plot representing a tuple of “(skills, open loop action sequences, skill-terminating states, and skill-terminating latent representations)”
    - There’s 3 types of arrows and they are all unlabeled in the figure, forcing us to use the caption which is 16 lines long.
    - Mutual information terms are mentioned but again without relation to an equation, I don’t know what’s going on
    - Overall, this figure could be much higher level representing the key intuitions of LPE.
- Notation: $\bar{a}$ to represent a vector of action sequences or something like this would be better than $a = [a_1, a_2,…]$
- Section 2.2 is far too long; each paragraph over-explains and can be much more concise. For example, the last paragraph essentially repeats the same statement multiple times regarding why Skillset Empowerment requires a dynamics model.
- Section 3.1: 3 lines of equations for LPE are directly presented without any intuition beforehand, this is difficult to parse. This section should be rewritten to instead introduce each part of the equation one at a time, before finally presenting the combined equation.
    - “The second term in the skillset diversity objective is an average KL divergence “… a simple reference to the specific equation number would be easier to parse.

**Minor issues:**

- L50: Eysenbach et al doesn’t require a model of transition dynamics, probably not a relevant citation here

**Results + Experiments:**

- Experiments don’t demonstrate the usefulness of the skillsets for downstream learning in terms of some downstream task reward, which is a huge con. Skills learned only matter if they are useful for some downstream application.
- All the baselines are based on Skill Empowerment. The authors should evaluate on additional modern skill discovery algorithms that don’t require simulators such as [CIC](https://arxiv.org/abs/2202.00161) or [METRA](https://arxiv.org/abs/2310.08887). It would also be useful to include the baselines in Skill Empowerment in the table since the environments are the same.

**Questions:**

Why are you only using $s_n$ for the mutual information maximization objective? Can’t a trivially diverse skill-space have very similar $n-1$ step trajectories that just result in a different $n$th state?

---

### Official Review · Reviewer_13Ps · 2024-10-30

**Soundness:** 2
**Presentation:** 1
**Contribution:** 1
**Rating:** 3
**Confidence:** 4

**Summary:**

This paper proposes a new algorithm named latent-predictive empowerment (LPE), which computes the empowerment defined by the mutual information between a latent random variable $Z$ and the final state random variable $S_n$ given the first state. LPE is built upon its previous work, Skillset Empowerment (SE) (Levy et al., 2024), which uses a DIAYN-like objective but with additional conditioning on $(\phi, \pi)$. The main differences between LPE and SE are (1) the use of a latent random variable $Z_n$ instead of the raw state $S_N$, and (2) the presence of an additional KL divergence term (the second term in $J(s_0, \phi, \pi)$ in Equation (6)). These changes enable computing empowerment without requiring a state dynamics model (but with a latent open-loop dynamics model). The authors show that this leads to better performance than SE and its variants in the stochastic four rooms, RGB QR code, and mountain car domains.

**Strengths:**

* Compared to its previous work SE, which requires a transition dynamics model $p(s' \mid s, a)$, the proposed method doesn't require it, and thus is relatively better scalable to high-dimensional image-based domains.

**Weaknesses:**

* It is unclear how good LPE is compared to other empowerment or unsupervised skill discovery methods. The authors only compare LPE with SE (the closest work) and its ablations, and the domains are mostly simple and a bit "contrived". For example, it is unclear how it works in more complex environments, such as Gym MuJoCo locomotion tasks, URLB tasks, robotic manipulation domains, or similar realistic environments, and how good it is compared to other existing unsupervised skill learning methods (like DIAYN (Eysenbach et al., 2018), DADS (Sharma et al., 2020), VGCRL (Choi et al., 2021), DISDAIN (Strouse et al., 2021), CIC (Laskin et al., 2022), METRA (Park et al., 2024), etc.). Without such results, it is difficult to assess the contributions of this new objective.
* The manuscript makes several questionable claims:
    * "recent empowerment approaches assume the agent has access to a model of the transition dynamics (i.e., a simulator of the environment) (Eysenbach et al., 2018; Gu et al., 2021; Levy et al., 2023; 2024)." I don't think the DIAYN objective (Eysenbach et al., 2018) assumes access to the ground-truth dynamics model. As far as I know, most of the existing mutual information skill learning methods do not require fitting a dynamics model (DADS (Sharma et al., 2020) is one notable exception). I believe the second sentence in the abstract as well as several claims in the paper are misleading, and should be revised.
    * "Skillset Empowerment was the first unsupervised skill learning algorithm to learn large skillsets in domains with stochastic and high-dimensional observations": This sentence is unscientific. What is the definition of "large" and in what sense is it the first unsupervised skill learning method to learn "large" skillsets in domains with stochastic and high-dimensional observations? I checked the domains used in Levy et al. 2024, and the didactic environments there really don't seem "large" enough to me. I'd recommend either cutting this sentence or revising it to an objective, justifiable claim.
* The paper is hard to read and does not flow well. For example, the main objective (Eq. 6) appears out of nowhere, without any motivation for the terms in the objective. They are (partly) explained in the following section, but several notations are not properly introduced, which makes it really difficult to read. For example, what are $Z_n$, $p_\eta$, and $p_\xi$? Where can I find the definition of $Z_n$? I recommend defining the notations before (or right after) the first occurrence (in this case, near Eq. (6)).
* The method requires fitting a latent *open-loop-action-conditioned* dynamics model $p(z_n \mid z, a_0, a_1, \ldots, a_{n-1})$. I'm not sure how scalable or tractable this is, especially in stochastic environments, because errors accumulate over time and the resulting open-loop-action-conditioned latent distribution can be arbitrarily complex. In contrast, many previous mutual information skill discovery methods (e.g., DIAYN) do not require fitting any (latent or state) dynamics model.

**Questions:**

Please refer to the weaknesses section above. Additionally, can the authors elaborate more on the rationale behind the second term in $J(s_0, \phi, \pi)$? Why do we need to have this specific KL divergence as a regularizer? Can this be justified as a consequence of a different, more principled objective instead of a rather "ad-hoc" explanation (L231-L243)?

---

### Official Review · Reviewer_D8n5 · 2024-10-31

**Soundness:** 3
**Presentation:** 2
**Contribution:** 3
**Rating:** 6
**Confidence:** 3

**Summary:**

The authors propose a novel algorithm LPE to learn skillsets of agents, without the access to transition dynamics. LPE simultaneously maximizes the diversity of skills and avoids similar terminal states resulting from different skills. Experimental results show the superiority of LPE in the environments in the fundamental baseline.

**Strengths:**

1. The proposed algorithm LPE sounds effective as a scalable method to compute the empowerment of states.
2. Experimental results show the superiority of LPE in the tasks from the fundamental baseline algorithm.

**Weaknesses:**

1. It seems that the environments in the experiments are still simple. Since LPE is proposed to function as a scalable method, it would be better to validate the effectiveness of LPE in more complex and practical environments, especially in tasks with higher-dimensional state spaces and more complex dynamics.
2. The authors are expected to improve writing for easier reading, especially in 3.1. Besides, there are some typos. For example, “defined” in Line 127 and “the” in Line 246 are repeated.

**Questions:**

See the weaknesses.

---

### Official Review · Reviewer_pQ6d · 2024-11-04

**Soundness:** 2
**Presentation:** 2
**Contribution:** 2
**Rating:** 3
**Confidence:** 4

**Summary:**

The paper proposes "Latent-Predictive Empowerment" (LPE), an empowerment-based algorithm designed to train agents to develop large, diverse skillsets without requiring a simulator of the environment. The method maximizes the mutual information between an agent’s skills and states while adding an additional KL term to learn the latent-predictive model. Experiments show that LPE performs comparably to empowerment methods that use a simulator.

**Strengths:**

The motivation of measure empowerment without requiring a full simulator of the environment’s transition dynamics makes sense to me.

**Weaknesses:**

1. The paper looks highly similar to the paper “Learning Large Skillsets in Stochastic Settings with Empowerment”. The whole paragraphs and equations, e.g. Background and Results, are the same with only a few words being different. Also, the paper is difficult to read and the narrative can be simplified and improved.
2. The experiments are toy examples. The baselines are not strong and comprehensive enough and the preference improvement is marginal. More environments and more complex tasks may be considered.
3. The proposed objective and the use of the latent predictive method is quite incremental. For me it is not well-motivated what is the importance of measuring "empowerment” to this field.
4. It seems the whole system relies heavily on the latent-predictive model. If the latent space does not effectively capture the diversity of possible states, it may struggle to distinguish between truly diverse skills and redundant ones. The authors may explain if there is a theoretical guarantee or any experiments results can demonstrate this concern.

**Questions:**

1. According to the weakness part, could authors provide more complicated higher-dimensional experimental results?
2. It would be great if the author could explain more about why “measuring empowerment” is an important topic and how it can contribute to downstream RL algorithms.
3. Following weakness 4, would it be possible for the authors to provide more analysis on the latent-predictive model?

**Details Of Ethics Concerns:**

The paper looks highly similar to the paper “Learning Large Skillsets in Stochastic Settings with Empowerment”. The whole paragraphs and equations, e.g. Background and Results, are the same with only a few words being different.

---

### Meta-Review · Area_Chair_RJGz · 2024-12-20

**Metareview:**

Most of the reviewers (three out of four) recommend rejecting this paper concerning its novelty, clarity (introduction, method, captions, etc.), dependence on a good latent open-loop-action-conditioned dynamics model, limited scenarios for evaluation (low-dimensional environments), and insufficient baselines, while the authors did not provide a rebuttal to address these issues. The only reviewer who initially felt positive about this paper (with a score of 6) decided not to champion the paper, considering the other reviews.  Consequently, I recommend rejecting the paper.

Additionally, I would like to urge the authors to at least show appreciation for the efforts the reviewers put into helping them improve the submission.

**Additional Comments On Reviewer Discussion:**

N/A: the authors have not provided a rebuttal.

---

### Decision · Program_Chairs · 2025-01-22

Reject